# β-catenin condensation facilitates clustering of the cadherin/catenin complex and formation of nascent cell-cell junctions

Jooske L. Monster [1,3], Caterina Manzato [1,3], Jan A. van der Beek [1], Willem-Jan Pannekoek[1], Janneke A. Hummelink[1], Michael A. Hadders [1,2], Cecilia de Heus[1], Judith Klumperman [1], Jurian Schuijers [1,2] ✉ & Martijn Gloerich [1] ✉

Cadherin-based junctions establish dynamically regulated adhesion between cells to coordinate tissue integrity and morphogenetic movements. Adhesion strength can be modulated by the organization of individual cadherin complexes into lateral clusters. Here, we identify a clustering mechanism of the cadherin complex established by its core component β-catenin. We show that the disordered termini of β-catenin drive the formation of condensates that incorporate other components of the cadherin complex in vitro. Using β-catenin mutants with hampered condensation, we demonstrate that β-catenin condensation nucleates the formation of submicron cadherin/catenin clusters that further develop into stable sites of adhesion. Furthermore, we show that β-catenin-dependent clustering ensures the efficient formation of de novo cell-cell adhesions. Our data thus indicate a role for β-catenin condensates in the supramolecular organization of the cadherin complex, and reveal that the function of β-catenin in the cadherin complex extends beyond connecting cadherin to α-catenin and the actin cytoskeleton.

Cadherin-based adherens junctions are integral to epithelial cohesion, by establishing cell-cell adhesion through cadherin *trans*-interactions between neighboring cells and intracellular connections with the actin cytoskeleton[1]. During development and homeostasis, epithelial cohesion must be tightly coordinated with morphogenetic movements, which requires dynamic regulation of the adhesive properties of cadherin junctions[2-4]. This can be achieved by regulating the organization of cadherin complexes into lateral clusters on the cell surface, increasing the avidity of intrinsically weak cadherin *trans*-interactions[5-8]. Across different cell types and species various cadherin clusters have been identified, ranging from nano- to microscale supramolecular structures (reviewed by refs. 9, 10). The formation of these clusters is mainly attributed to *cis*-interactions between ecto-domains of E-cadherin and association with the actin cytoskeleton that, for instance, limit lateral diffusion[11-18]. Nonetheless, the molecular

mechanisms regulating clustering, including the contribution of other cadherin-complex components, is incompletely understood[10]. Moreover, how clustering of the cadherin complex impacts adhesion dynamics in epithelial tissues remains to be answered.

Over the last years it has become clear that clustering of proteins at the membrane, including transmembrane receptor complexes, is facilitated by liquid-liquid phase separation[19-21]. This biophysical process concentrates proteins into biomolecular condensates through weak multivalent interactions, for instance between intrinsically disordered regions (IDRs) enriched for polar, charged, or aromatic amino acids[22-24]. Biomolecular condensates typically exhibit liquid-like properties, and by locally concentrating proteins, these dynamic structures can catalyze enzymatic reactions, sequester molecules, or serve as a structural organization platform[25,26]. The formation of condensates has recently been implicated in the assembly and regulation of cell

[1]Center for Molecular Medicine, University Medical Center Utrecht, Utrecht University, Utrecht, The Netherlands. [2]Oncode Institute, Utrecht, The Netherlands. [3]These authors contributed equally: Jooske L. Monster, Caterina Manzato. ✉e-mail: J.Schuijers@umcutrecht.nl; M.Gloerich@umcutrecht.nl

adhesion complexes[27]. The tight junction (TJ) scaffolding zonula occludens (ZO) proteins form biomolecular condensates and thereby organize into membrane-attached compartments that locally enrich other components of this complex during formation of nascent TJs[28,29]. Multiple focal adhesion (FA) proteins exhibit condensation behavior, which contributes to integrin clustering, and the assembly and dynamics of FAs[30–32]. Whether formation of biomolecular condensates also contributes to the supramolecular organization of other adhesion complexes, including cadherin-based adhesions, remains to be determined.

β-Catenin is a central component of the cadherin complex[1]. It binds to E-cadherin immediately following synthesis of E-cadherin at the membrane of the endoplasmic reticulum, and establishes the link with α-catenin and consequently the actin cytoskeleton after their arrival at the plasma membrane[33,34]. The cellular function of β-catenin extends beyond cell-cell adhesion, as it also acts as transcriptional transactivator in the Wnt-signaling pathway[35]. The transcriptional role of β-catenin was recently shown to be supported by its capability to form biomolecular condensates, relying on weak intermolecular interactions through its structurally disordered N- and C-termini[36–38]. β-Catenin partitions into condensates at transcriptional enhancer elements, thereby promoting transcription of Wnt-target genes[36]. In this study, we investigated whether the condensation of β-catenin also impacts the formation and organization of cell-cell junctions. We find that β-catenin forms phase-separated droplets in vitro in the presence of E-cadherin and α-catenin and co-partitions these other cadherin complex components into condensates. Using β-catenin mutants with impaired condensate formation capacity, we demonstrate that β-catenin condensation promotes its clustering together with E-cadherin and α-catenin at the cell cortex and facilitates the formation of nascent cell-cell junctions. Our data thus indicate a role for β-catenin condensates in the supramolecular organization of the cadherin complex, and provide evidence for the importance of cadherin-complex clustering in establishing adhesion between two opposing cells.

## Results

### β-Catenin co-partitions with other components of the cadherin complex in vitro

The contribution of β-catenin condensate formation to its transcriptional function[36] raises the question whether and how the condensation of β-catenin is linked to its function in cell-cell adhesion complexes (Fig. 1A, left). Incorporation in the cadherin complex through interactions with E-cadherin and α-catenin could impair β-catenin condensate formation, or alternatively, β-catenin may integrate these other components of the cadherin complex into condensates. To test these possibilities, we analyzed the formation of phase-separated droplets of purified β-catenin in vitro in the presence of E-cadherin and α-catenin, using recombinant β-catenin, the cytosolic tail of E-cadherin, and α-catenin, conjugated with fluorescent proteins (mEGFP-β-catenin$^{Wt}$, E-cadherin$^{cyto}$-mTagBFP2, and mCherry-α-catenin, respectively) (Fig. 1A, right).

We and others previously showed that upon induction of crowding by addition of 10% PEG-8000, mEGFP-β-catenin partitioned into spherical, micron-sized droplets in a concentration-dependent manner (refs. [36–38], Figs. 1B, C, and S1B). Under similar conditions, E-cadherin$^{cyto}$-mTagBFP2 did not form droplets (Figs. 1B, C and S1B). However, when combined at equimolar levels, similar to their stoichiometry at the plasma membrane in cells[33,39], E-cadherin$^{cyto}$-mTagBFP2 incorporated into droplets of mEGFP-β-catenin (Fig. 1C, D). α-Catenin contains predicted IDRs (Fig. S1A), and formed droplets by itself at comparable concentrations as mEGFP-β-catenin (Figs. 1B, C and S1B). Furthermore, mCherry-α-catenin co-partitioned with β-catenin, both in the absence and presence of E-cadherin (Fig. 1C, D, and F). In contrast, unconjugated mTagBFP2 and mCherry controls were not able to form droplets nor partition into β-catenin droplets

(Figs. 1C, D and S1C). Live-imaging revealed fusion of proximal droplets containing β-catenin, E-cadherin$^{cyto}$, and α-catenin, supporting their liquid-like properties (Fig. 1E and Movies S1 and 2). Altogether, these data demonstrate that β-catenin forms phase separated droplets in the presence of E-cadherin and α-catenin and these other cadherin complex components co-partition into these droplets.

We previously identified that condensate formation of β-catenin relies on weak intermolecular interactions through aromatic amino acids in its N- and C-terminal IDRs[36–38], which do not overlap with the established high-affinity binding sites in β-catenin for E-cadherin (Fig. S1D)[40–44]. Mutation of these aromatic residues (from here on referred to as β-catenin$^{IDRs*}$) diminished the partitioning of β-catenin, as well as co-partitioning of E-cadherin and α-catenin (Figs. 1F, G and S1B and E). The small fraction of β-catenin$^{IDRs*}$ droplets that did form contained all three proteins, indicating that the mutations impair the ability of β-catenin to form droplets but not its interactions with E-cadherin and α-catenin (Fig. 1G). These findings indicate that condensation of β-catenin drives the concentration of all three core cadherin complex components into droplets. In line with this, even though β-catenin and α-catenin both form droplets at similar concentrations, α-catenin did not co-partition with E-cadherin$^{cyto}$ in the absence of β-catenin (Fig. 1C, D). Altogether, these data demonstrate that E-cadherin and α-catenin co-partition with β-catenin into phase-separated droplets formed by the IDR-dependent condensation of β-catenin.

### IDR-dependent clustering of β-catenin at the cell cortex

Having established that β-catenin condensation co-partitions E-cadherin and α-catenin into droplets in vitro, we made use of β-catenin$^{IDRs*}$ with hampered condensation to investigate whether this behavior contributes to its organization at cell-cell contacts. Live-cell imaging of endogenously mEGFP-tagged β-catenin in the colorectal cancer cell line HCT116, mouse embryonic stem cells, and embryonic kidney cell line HEK293T displayed a distinct organization of β-catenin into small clusters (~250–600 nm in diameter) along the cell cortex (Figs. 2A and S2A, B and Movies S3, 5, and 6). Further characterization of HCT116 cells at varying cell densities revealed that mEGFP-β-catenin cluster formation was not limited to cell-cell contacts, but also occurred at free membranes lacking neighboring cell contacts (Fig. 2A and Movie S3). Moreover, we occasionally observed the coalescence of individual cortical mEGFP-β-catenin clusters (Fig. S2C and Movie S7).

We used the HCT116 cell line to test whether β-catenin condensation contributes to the formation of these clusters. For this, we generated β-catenin knock-out (β-catenin$^{KO}$) lines of HCT116 and re-introduced either wildtype mScarlet-β-catenin (mSc-β-catenin$^{Wt}$) or mScarlet-β-catenin with mutated IDRs (mSc-β-catenin$^{IDRs*}$; Figs. 2B and S2D). Both β-catenin variants were similarly able to localize to cell-cell contacts (Fig. S2E), in line with the notion that the IDRs are not required for β-catenin to bind E-cadherin (Fig. S1D)[40–42]. However, the presence of the distinct, small clusters at the cell cortex observed for β-catenin$^{Wt}$ was strongly reduced in cells expressing β-catenin$^{IDRs*}$ (Fig. 2C–E). Moreover, exogenously expressed mSc-β-catenin$^{IDRs*}$ did not incorporate into clusters formed in cells expressing endogenous β-catenin conjugated to mEGFP, whereas the distribution of mSc-β-catenin$^{Wt}$ closely overlapped with the clusters formed by endogenous mEGFP-β-catenin (Fig. 2F–H). Similarly, clusters of endogenous mEGFP-β-catenin at the free membrane selectively incorporated wildtype β-catenin and not β-catenin$^{IDRs*}$ (Fig. S2F). Thus, the IDRs of β-catenin are essential for the formation of β-catenin clusters at the cell cortex, and β-catenin with hampered condensation capacity fails to integrate into existing clusters of wildtype β-catenin.

To corroborate our findings indicating a role for β-catenin condensation in its clustering at the cell cortex and exclude potential confounding effects of the double-mutated IDRs, we designed additional β-catenin mutants with impaired condensate formation capacity. We previously showed that loss of a single β-catenin IDR severely affects the

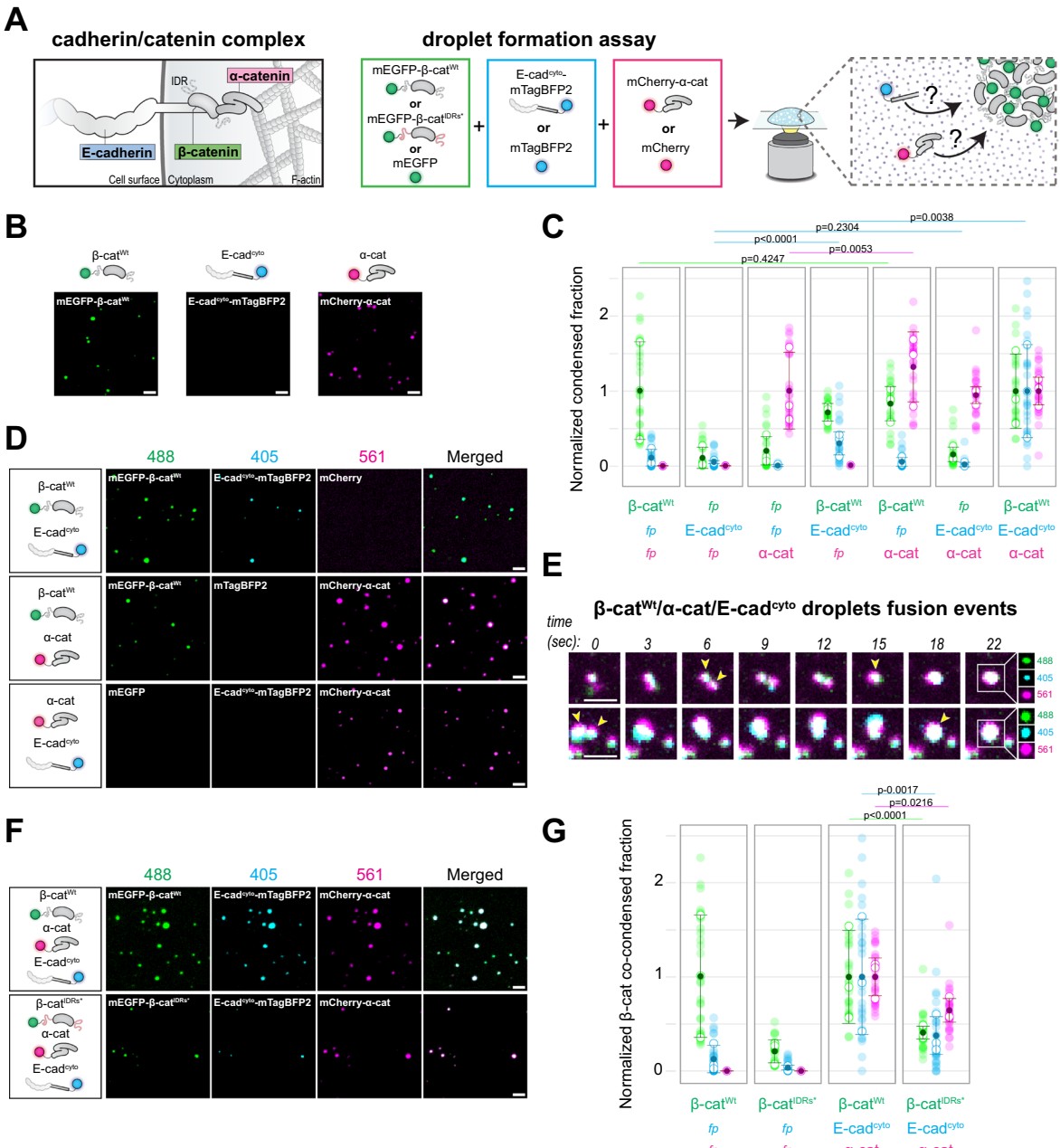

**Fig. 1 | Droplet formation assay of β-catenin and components of the cadherin complex in vitro. A** Left: Schematic representation of E-cadherin (E-cad), β-catenin (β-cat), α-catenin (α-cat), and F-actin in cadherin-based adherens junctions. Right: Schematic representation of the experimental procedure of the droplet formation assay. Purified recombinant proteins (mEGFP-β-cat^Wt, mEGFP-β-cat^IDRs*, E-cad^cyto-mTagBFP2 or mCherry-α-cat), or the unconjugated fluorescent proteins (FPs) as control, were mixed together in equimolar concentrations (1 μM) and the formation of droplets upon addition of 10% PEG-8000 was microscopically analyzed. **B** Representative images of droplet assay of purified mEGFP-β-cat^Wt, E-cad^cyto-mTagBFP2, or mCherry-α-cat (in the presence of the unconjugated FPs as controls, not shown). **C** Quantification of the condensed fraction of purified cadherin/catenin complex components or unconjugated FP controls. The condensed fraction indicates the percentage of fluorescent intensity inside droplets over the total intensity. Data was normalized to the mEGFP-β-cat^Wt/mCherry-α-ca/E-cad^cyto-mTagBFP2 condition for each channel separately. Each semi-transparent data point represents an image (*n* = 10), each white datapoint represents mean of independent replicates (*N* = 3), full darker point indicates the mean of three experiments.

**D** Representative images of droplet co-partition assay of the indicated purified cadherin/catenin complex components, or their unconjugated FP counterparts as controls. **E** Representative still images of fusion events of droplets containing mEGFP-β-cat^Wt (488), mCherry-α-cat (561), and E-cad^cyto-mTagBFP2 (405). **F** Representative images of droplet co-partition assay of the indicated purified cadherin/catenin complex components. **G** Quantification of the fraction of purified cadherin/catenin complex components or unconjugated FPs co-condensed in mEGFP-β-cat droplets, indicating the percentage of fluorescent intensity inside mEGFP-β-cat^Wt droplets over the total intensity of the image. Data was normalized to the mEGFP-β-cat^Wt/mCherry-α-cat/E-cad^cyto-mTagBFP2 condition for each channel separately. Each semi-transparent data point represents an image (*n* = 10), each white data point represents mean of independent replicates (*N* = 3), full darker point indicates the mean of three experiments. For all graphs error bars indicate standard deviation between the different independent replicates, *P*-values are indicated, Two-tailed Kruskal–Wallis chi-squared test and Dunn's multiple comparison test. All scale bars represent 2 μm. Source data are provided as a Source Data file.

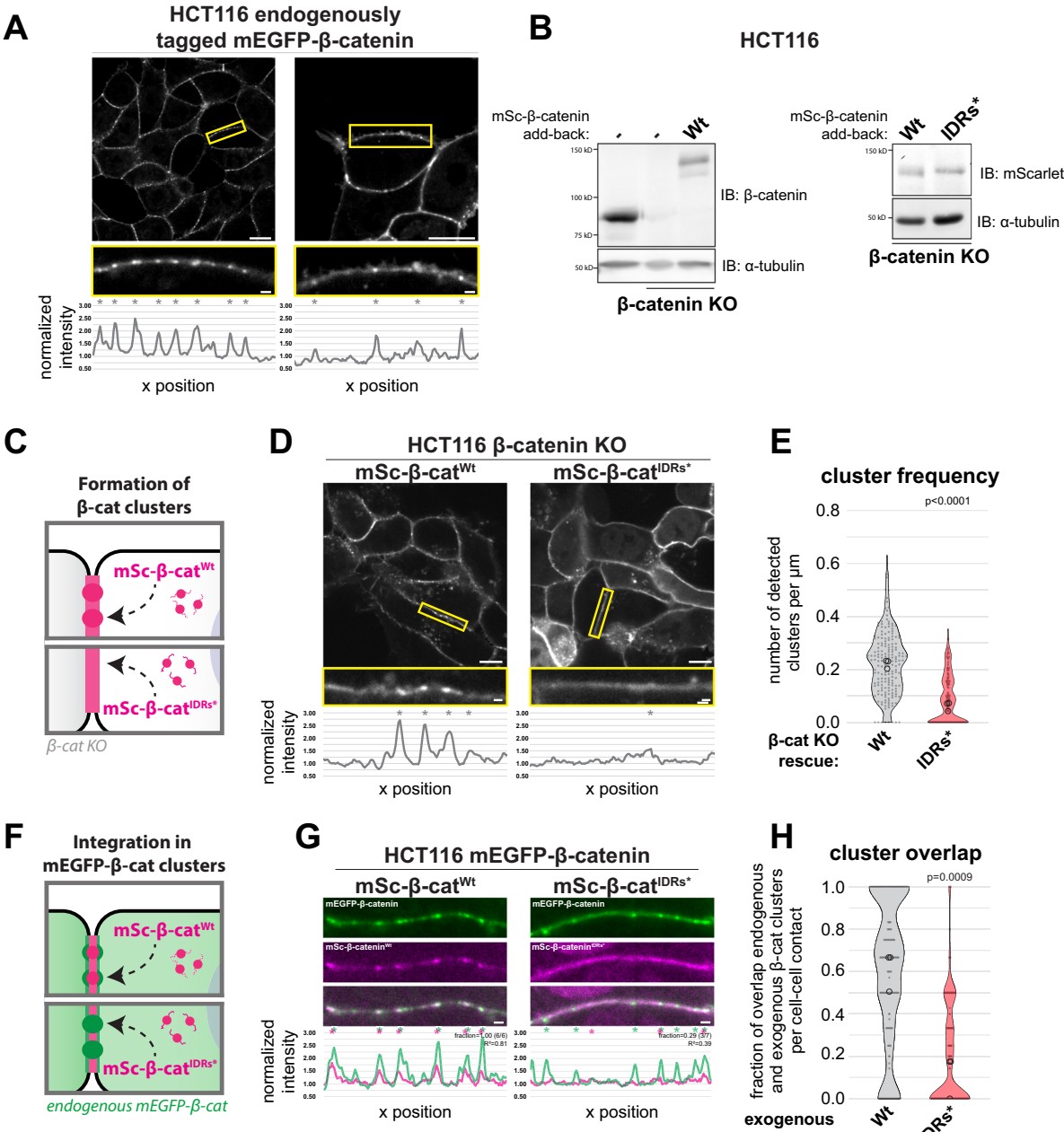

**Fig. 2 | Formation of β-catenin clusters at the cell cortex is dependent on its IDRs. A** Representative confocal images of endogenously tagged mEGFP-β-catenin in HCT116 cells at cell-cell contacts and at the free membrane, with zoom-ins (yellow box) of individual cell-cell contacts. Individual clusters are detected by peak detection on the corresponding line profile of the normalized fluorescent intensity of the junction, indicated by asterisks. See Movies S3 and 4 for z-stacks of the displayed images. **B** Left: Western blot of lysates of parental and β-catenin knock-out (KO) HCT116 cells and KO cells with addback of mSc-β-catenin^Wt, probed for β-catenin and α-tubulin. Right: Western blot of lysates of β-catenin KO HCT116 cells with addback mSc-β-catenin^Wt or mSc-β-catenin^IDRs* probed for mScarlet and α-tubulin. mScarlet antibody was used to compare the expression levels of exogenously expressed wildtype and mutated β-catenin to exclude potential effects of the introduced mutations in β-catenin on antibody affinity. **C** Schematic representation of assay to test β-catenin cluster formation of mSc-β-catenin^Wt and mSc-β-catenin^IDRs* at cell-cell contacts in β-catenin KO cells. **D** Representative confocal images of β-catenin KO HCT116 cells with addback of mSc-β-catenin^Wt or mSc-β-catenin^IDRs*. Zoom-in shows individual cell-cell contact and its corresponding line profile of the normalized fluorescent intensity and detected peaks (asterisks). Note that while one of the endogenous β-catenin alleles of HCT116 cells contains a S45 deletion disrupting destruction complex-mediated degradation, the reintroduced wildtype β-catenin does not, indicating that this mutation is not required for cortical β-catenin cluster formation. **E** Quantification of the number of detected peaks per μm of mSc-β-catenin^Wt or mSc-β-catenin^IDRs* expressed in β-catenin KO HCT116 cells. Each point represents a measurement for a single cell-cell contact ($n = 181$ and 186), pooled from three independent measurements (medians of each indicated with circles). **F** Schematic representation of assay to test integration of exogenously expressed mSc-β-catenin^Wt or mSc-β-catenin^IDRs* into mEGFP-β-catenin clusters of endogenously tagged HCT116 cells. **G** Representative confocal images of individual cell-cell contacts of HCT116 cells expressing both endogenously tagged mEGFP-β-catenin (green) and exogenously expressed mSc-β-catenin^Wt or mSc-β-catenin^IDRs* (magenta). The corresponding line profiles show the normalized fluorescent intensity and detected peaks (asterisks) of both fluorophores. The fraction of peaks detected in the mEGFP channel with an overlapping peak in the mScarlet channel and the Pearson correlation ($R^2$) between the line profiles of the two fluorophores are shown. **H** Quantification of the fraction of mEGFP-β-catenin peaks with an overlapping peak for exogenously expressed mSc-β-catenin^Wt or mSc-β-catenin^IDRs*, respectively, per cell-cell contact. Each point represents a measurement for a single cell-cell contact ($n = 161$ and 185), pooled from three independent measurements (medians of each indicated with circles). For all graphs *P*-values are indicated, nested two-sided *t*-test. Scale bars represent 10 μm (overview images) or 1 μm (zoom-ins of individual cell-cell contacts). Source data are provided as a Source Data file.

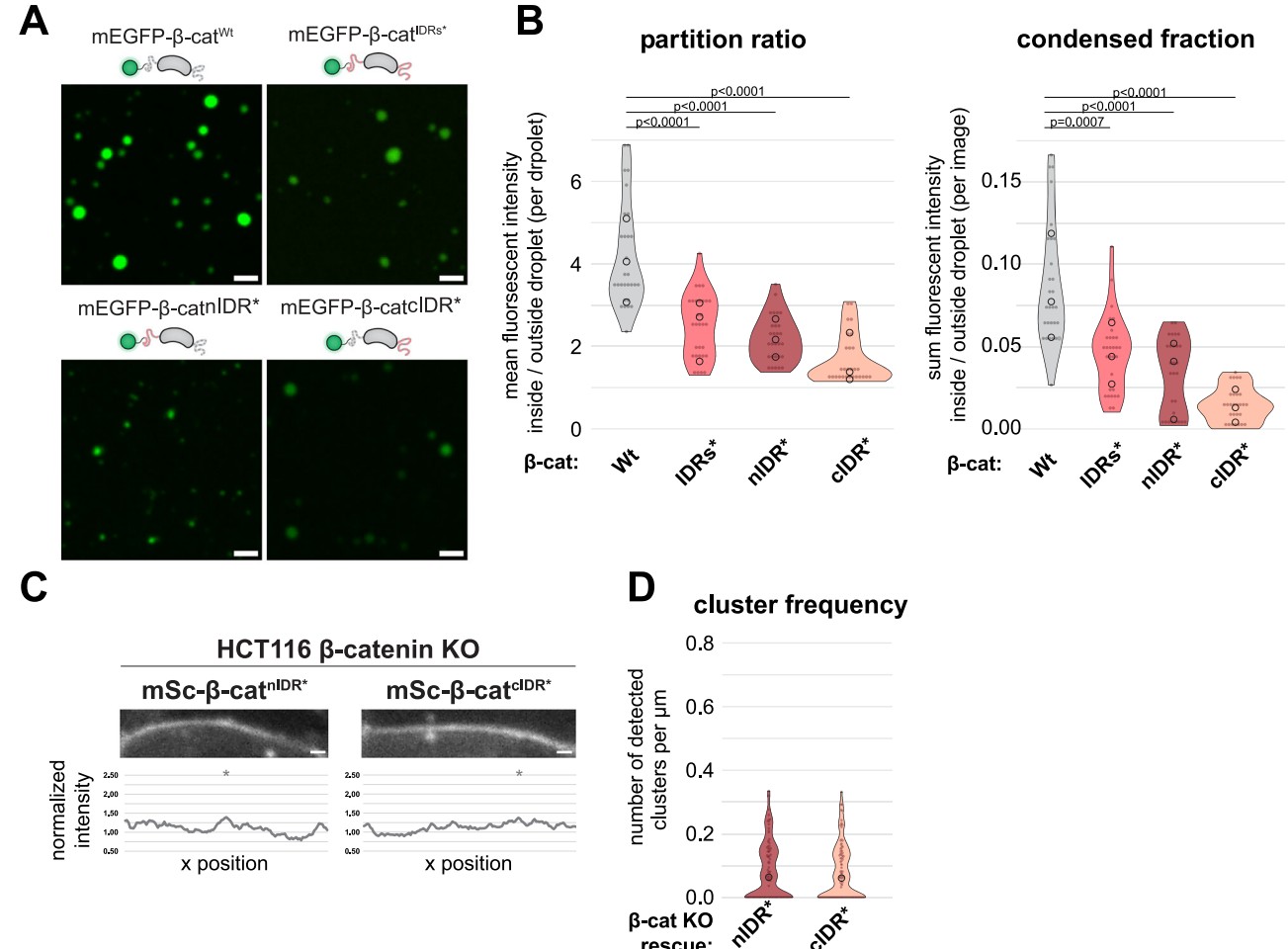

**Fig. 3 | Independent β-catenin IDR-mutants corroborate the importance of β-catenin condensation in cortical cluster formation. A** Representative confocal images of droplet formation assays of purified mEGFP-β-catenin^Wt, mEGFP-β-catenin^IDRs*, mEGFP-β-catenin^nIDR*, or mEGFP-β-catenin^cIDR* in the presence of 10% PEG-8000. **B** Quantification of the partition ratio and condensed fraction of purified mEGFP-β-catenin^Wt, mEGFP-β-catenin^IDRs*, mEGFP-β-catenin^nIDR* or mEGFP-β-catenin^cIDR* (8 µM). The partition ratio represents the mean intensity inside over outside droplets, while the condensed fraction represents the percentage of fluorescent intensity inside droplets over the total intensity of the image. Each point represents a measurement from one image ($n > 8$), pooled from three independent replicates (medians of each indicated with circles). *P*-values are indicated;

Two-tailed Kruskal–Wallis chi-squared test and Dunn's multiple comparison test. **C** Representative confocal images of β-catenin KO HCT116 cells with addback of mSc-β-catenin^nIDR* or mSc-β-catenin^cIDR*, showing individual cell-cell contact and the corresponding line profile of normalized fluorescent intensity and detected peaks (asterisks). **D** Quantification of the number of detected peaks per µm of mSc-β-catenin^nIDR* or mSc-β-catenin^cIDR* expressed in β-catenin KO HCT116 cells. The median is indicated with a circle and each point represents a measurement for a single cell-cell contact ($n = 92$ and 104). All scale bars represent 2 µm (droplet assay, **A**) or 1 µm (individual cell-cell contacts, **C**). Source data are provided as a Source Data file.

formation of droplets in vitro, implying both the N- and C-terminal IDR are essential for β-catenin condensate formation[36]. We mutated the aromatic amino acids in only the N- or C-terminal IDR of β-catenin (referred to as β-catenin^nIDR* and β-catenin^cIDR*, respectively) and both were sufficient to attenuate the formation of droplets (Fig. 3A, B). When these mutants were introduced into β-catenin^KO HCT116 cells (Fig. S3A), we found that neither of the mutants efficiently formed clusters at the cell cortex (Fig. 3C, D), similarly to β-catenin with both IDRs mutated (Fig. 2D, E). These mutants independently link an inhibition of condensate formation in vitro to the absence of cluster formation in cells. Thereby, these mutants corroborate the conclusion that β-catenin condensation contributes to its clustering at the cell cortex.

## IDR-dependent β-catenin clusters integrate other components of the cadherin/catenin complex to form stable adhesion complexes

Next, we aimed to investigate whether IDR-dependent clustering of β-catenin can drive the clustering of other components of the cadherin/

catenin complex. To this end, we first co-expressed fluorescently tagged versions of E-cadherin and α-catenin in HCT116 mEGFP-β-catenin cells. E-cadherin-mCherry and mCherry-α-catenin colocalized within the IDR-dependent mEGFP-β-catenin clusters at the plasma membrane (Fig. 4A–C). In contrast, mCherry non-specifically targeted to the plasma membrane by fusion to a C-terminal plasma membrane localization signal (CAAX-motif of K-Ras, see "Methods"), was not enriched at mEGFP-β-catenin clusters (Fig. 4A–C). Incorporation of E-cadherin into the IDR-dependent mEGFP-β-catenin clusters did not rely on its ability to form trans- or cis-interactions, as N-terminally truncated E-cadherin lacking these interaction regions was similarly integrated into mEGFP-β-catenin clusters (Fig. S4A). We further excluded that cortical β-catenin clusters were part of the β-catenin destruction complex, which mediates β-catenin degradation and was previously shown to assemble into supramolecular clusters at the cell cortex in other model systems[45,46]. However, the destruction complex component Axin1 was not detected in the β-catenin clusters at the cell cortex in HCT116 cells (Fig. S4B). Together, these findings demonstrate that

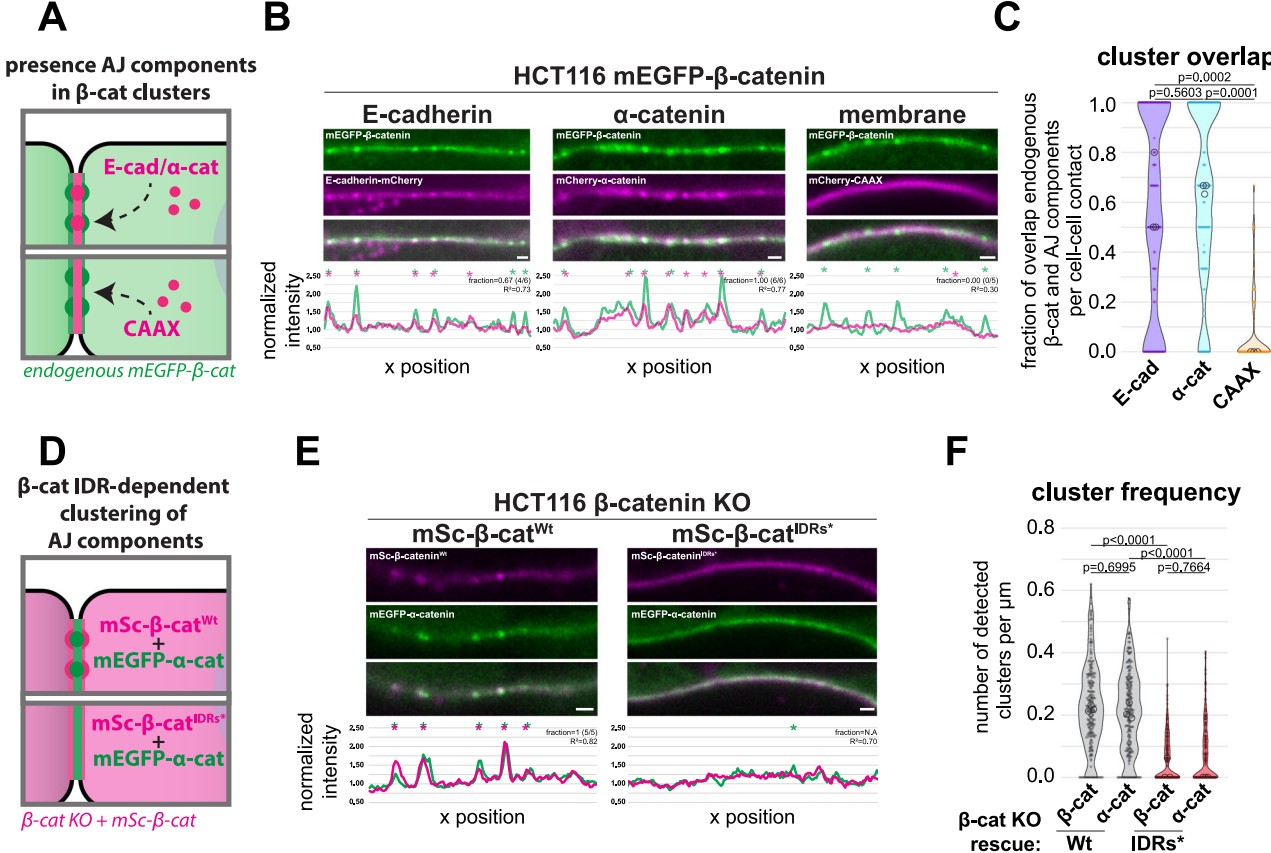

**Fig. 4 | IDR-dependent β-catenin clusters adherens junction components.**
**A** Schematic representation of assay to test enrichment of cadherin complex components (top) and the plasma membrane (visualized with mCherry-CAAX, bottom) in clusters of endogenously tagged mEGFP-β-catenin in HCT116 cells. **B** Representative confocal images of individual cell-cell contacts of HCT116 cells expressing both endogenously tagged mEGFP-β-catenin (green) and exogenously expressed E-cadherin-mCherry, mCherry-α-catenin, or mCherry-CAAX (magenta). The corresponding line profiles show the normalized fluorescent intensity and detected peaks (asterisks) of both fluorophores. The fraction of peaks detected in the mEGFP channel with an overlapping peak in the mScarlet channel and the Pearson correlation ($R^2$) between the line profiles of the two fluorophores are shown. **C** Quantification of the fraction of mEGFP-β-catenin peaks with an overlapping peak for exogenously expressed E-cadherin-mCherry (E-cad), mCherry-α-catenin (α-cat), or mCherry-CAAX (CAAX), per cell-cell contact. Each point represents a measurement for a single cell-cell contact ($n = 115, 128$, and $140$, respectively), pooled from three independent measurements (medians of each indicated

with circles). **D** Schematic representation of assay to test α-catenin clustering in β-catenin knock-out (KO) HCT116 cells with addback of mSc-β-catenin[Wt] or mSc-β-catenin[IDRs*], respectively. **E** Representative confocal images of individual cell-cell contacts of β-catenin KO HCT116 cells with addback of mSc-β-catenin[Wt] or mSc-β-catenin[IDRs*] (magenta), respectively, coexpressing mEGFP-α-catenin (green). The corresponding line profiles show the normalized fluorescent intensity and detected peaks (asterisks) of both fluorophores. The fraction of peaks detected in the mScarlet channel with an overlapping peak in the mEGFP channel and $R^2$ between the line profiles of the two fluorophores are shown. N.A not applicable, dividing by 0 peaks. **F** Quantification of the number of detected peaks of mEGFP-α-catenin (α-cat) and mS-β-catenin (β-cat) per μm, in β-catenin KO HCT116 cells with addback of wildtype (Wt) or IDRs-mutated (IDRs*) β-catenin. Each point represents a measurement for a single cell-cell contact ($n = 222$ and $213$), pooled from three independent measurements (medians of each indicated with circles). For all graphs P-values are indicated, nested one-way ANOVA Tukey's multiple comparisons test. All scale bars represent 1 μm. Source data are provided as a Source Data file.

the IDR-dependent mEGFP-β-catenin clusters represent clustered cadherin complexes. Next, we tested whether β-catenin condensate formation is essential for the clustering of the other cadherin complex components, or if these instead can independently form clusters. We therefore exogenously expressed mEGFP-α-catenin in β-catenin[KO] HCT116 cells rescued with either wildtype or the IDRs-mutated mSc-β-catenin, which showed that the α-catenin clusters were not formed in the presence of β-catenin[IDRs*] (Fig. 4D–F). Altogether, our data show that β-catenin condensation clusters other cadherin complex components and thereby contributes to their supramolecular organization at cell-cell contacts.

To further examine IDR-dependent β-catenin clusters, we applied electron microscopy (EM). We prepared 90–100 nm cryosections of mEGFP-β-catenin HCT116 cell monolayers flat-embedded to preserve the orientation of the cells[47]. Endogenous mEGFP-β-catenin expression was too low to discern clear sites of enrichment by immunogold labeling of GFP. To circumvent this, we imaged the same samples by correlative light-EM (CLEM) (Figs. 5A and S5A). Highly sensitive

fluorescence microscopy revealed mEGFP signal was preserved and well visible, even in ultrathin sections. Hence, we used mEGFP fluorescence to guide us to β-catenin labeled regions by light microscopy and sequentially imaged the same regions by EM. By overlaying the two images we discerned that clusters of mEGFP-β-catenin identified by light microscopy represent cell-cell junctions, typically of a focal nature, with high protein densities at the adjacent cytosolic sites (Figs. 5A and S5A). The overall cell morphology and that of the identified cell-cell adhesions was similar in classical resin-embedded EM, validating our CLEM findings (Fig. S5B)[48–50]. These results indicate that the IDR-dependent β-catenin clusters at cell-cell contacts represent sites of intercellular adhesion. In line with this, mosaic cultures of mEGFP-β-catenin cells and mSc-β-catenin cells revealed that the IDR-dependent β-catenin clusters integrated β-catenin from both cells forming the cell-cell contact, and thus appear to be part of intercellularly connected adhesion complexes (Fig. 5B, C). Furthermore, disruption of the actin network by addition of Cytochalasin D altered the organization of the β-catenin clusters, with individual clusters

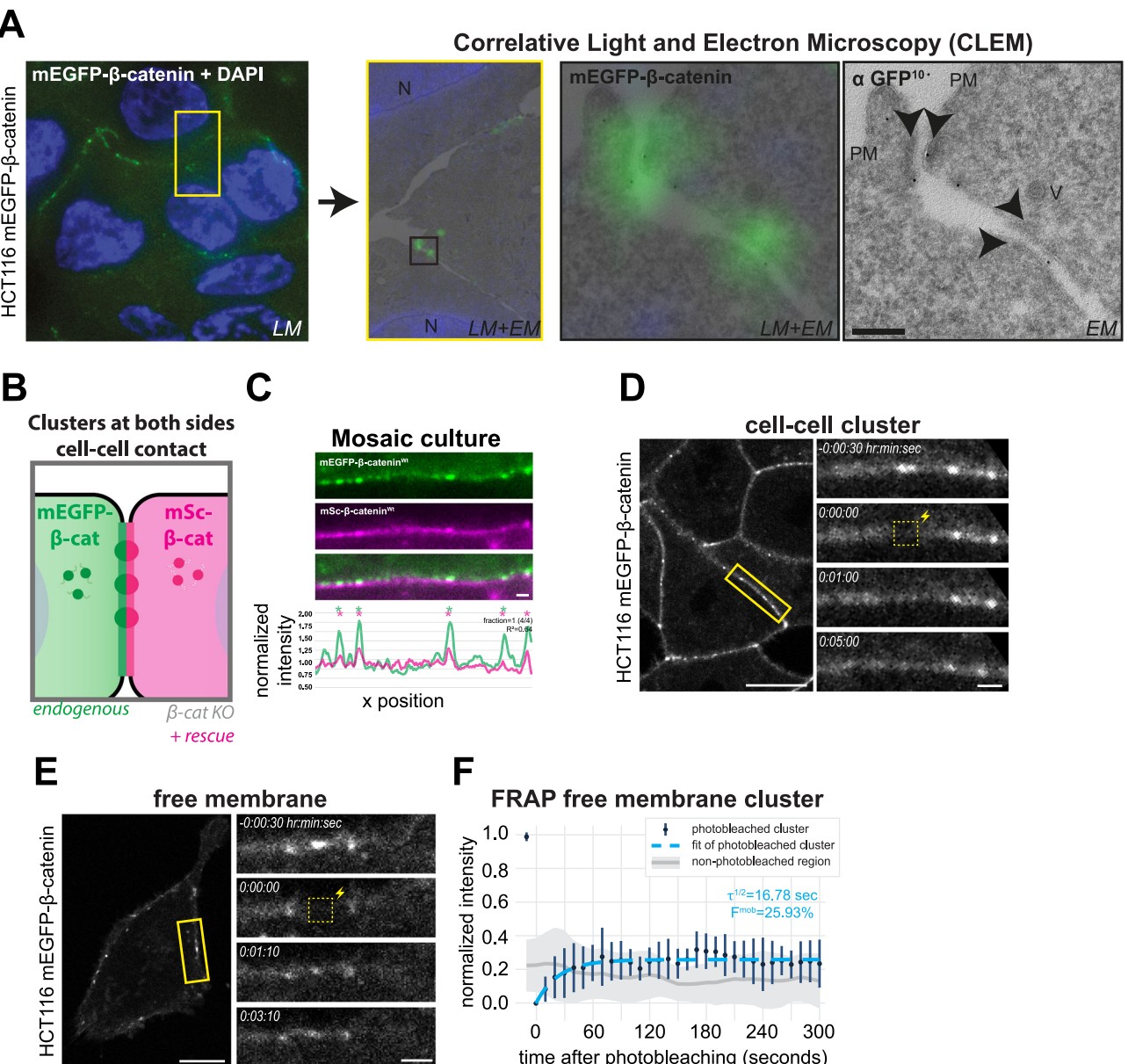

**Fig. 5 | Characterization of β-catenin clusters at the free membrane and cell-cell contacts. A** Representative example of correlative Light and Electron Microscopy (CLEM) image of HCT116 cells with endogenously tagged mEGFP-β-catenin in 90–100 nm thawed cryosections. Light-microscopy (LM) images of mEGFP-β-catenin (green) together with DAPI (blue) were overlayed with EM images with immuno-EM labeling of mEGFP with 10 nm gold particles. Sites of punctuated mEGFP-β-catenin enrichment typically coincided with the close proximity of membranes of two adjacent cells and an intracellular electron density (arrowheads). Note that the correlative GFP fluorescence corresponds to sites with immunogold labeling for GFP, aiding the detection and characterization of these sites. For additional examples, see Fig. S5A. The analysis involved 4 individual samples from 2 biological replicates. N nucleus, PM plasma membrane, V vesicle. **B** Schematic representation of mosaic cultures of HCT116 cells expressing either endogenously tagged mEGFP-β-catenin, or mSc-β-catenin[Wt] in a β-catenin KO background. **C** Representative confocal image of an individual cell-cell contact in a mosaic culture of HCT116 cells expressing either endogenously tagged mEGFP-β-catenin (green, top cell) or mSc-β-catenin[Wt] in a β-catenin KO background (magenta, bottom cell). The corresponding line profile shows the normalized fluorescent intensity and detected peaks (asterisks) of both fluorophores in the same colors. The fraction of peaks detected in the mEGFP channel with an overlapping peak in the mScarlet channel and the Pearson correlation ($R^2$) between the line profiles of the two fluorophores are shown. **D** Representative still images of Fluorescent Recovery After Photobleaching (FRAP) experiment (see Movie S9). A single mEGFP-β-catenin cluster at a cell-cell contact of endogenously tagged HCT116 cells was photobleached (yellow box, time point 0) and imaged over time. See Fig. S5D for the quantification of recovery over time. **E** Representative still images of FRAP experiment. A single mEGFP-β-catenin cluster at the free membrane of endogenously tagged HCT116 cells was photobleached (yellow box, time point 0) and imaged over time (see Movie S10). **F** Quantification of the normalized fluorescent signal (mean ± SD) before and after photobleaching of mEGFP-β-catenin clusters at the free membrane ($n = 10$). Normalized fluorescent intensities of single clusters were measured every 10 s (blue dots), and the measurements were fitted to a single exponential curve (blue dotted line ±SD) to calculate the mobile fraction ($F^{mob}$) and the half-life recovery time ($\tau^{1/2}$; in seconds). Intensities of a distant region of the junction was similarly measured in each frame (gray line ±SD), to compare the recovered intensity of the clusters to the intensity level of the linear junction. Scale bars represent 10 µm (overview images **D**, **E**), 1 µm (single cell-cell contacts; **C–E**), or 200 nm (EM image **A**). Source data are provided as a Source Data file.

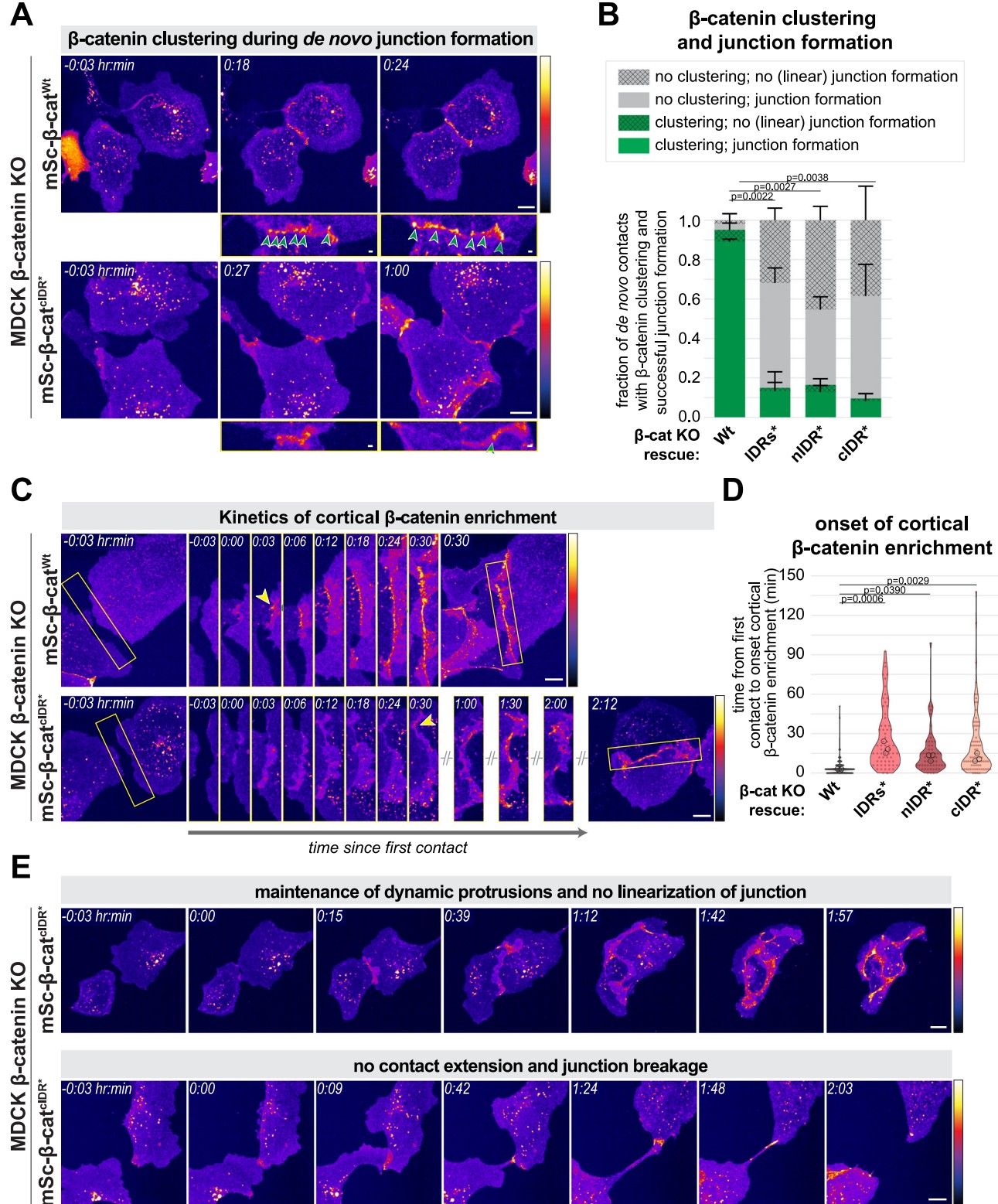

coalescing into larger structures, which supports the idea that these clusters are engaged with the actin network (Fig. S5C and Movie S8).

The identification of cadherin/catenin clusters relying on β-catenin condensation provides a conundrum, as biomolecular condensates typically behave as dynamic, liquid-like structures that appear to be antithetical with the role of cadherin adhesions as force-bearing structures[25,26]. We therefore assessed the dynamics of β-catenin in the IDR-dependent clusters using fluorescence recovery after photobleaching (FRAP). This showed that mEGFP-β-catenin

clusters at cell-cell contacts did not recover beyond the fluorescence levels of the surrounding membrane outside of the clusters, indicating that the observed IDR-dependent clusters at cell-cell contacts are immobile, stable structures (Figs. 5D and S5D and Movie S9). In contrast, photobleached clusters at the free membrane exhibited fluorescence recovery within the cluster, and showed recovery kinetics of $t_{1/2} = 17$ s (Fig. 5E, F and Movie S10). These findings indicate the existence of dynamic IDR-dependent E-cadherin/β-catenin clusters at the cell cortex outside of cell-cell contacts. Altogether, our findings support a

**Fig. 6 | β-Catenin-dependent clustering facilitates junction formation in MDCKs. A** Representative still images (Movies S11 and S12) with zoom-in (bottom) of cluster formation (green arrowheads) during de novo contact formation between two β-catenin knock-out (KO) MDCK cells with addback of mSc-β-catenin[Wt] or mSc-β-catenin[cIDR*] (Fire LUT). Time point 0 marks initial contact formation. For KO cells with addback of mSc-β-catenin[IDRs*] or mSc-β-catenin[nIDR*], see Movies S13 and S14, respectively. **B** Quantification of the fraction of de novo cell-cell contacts in β-catenin KO MDCK cells with mSc-β-catenin[Wt], mSc-β-catenin[IDRs*], mSc-β-catenin[nIDR*] or mSc-β-catenin[cIDR*] addback, showing clustering (green) or no clustering (grey) and ability to form a linear junction (indicated with cross-hatching). Bars indicate mean fraction ± SD from three independent experiments ($n = 91$, 64, 55, and 73 contacts in total). **C** Representative time-lapse images (Movies S15 and S16) of the formation of a de novo cell-cell contact (zoom-in; yellow box) between two β-catenin KO MDCK cells with mSc-β-catenin[Wt] or mSc-β-catenin[cIDR*] addback (Fire LUT). The onset of cortical β-catenin enrichment (marked by yellow arrowhead) and the completion of a linear contact (final frame) is delayed

in cells expressing mSc-β-catenin[cIDR*] (shown as representative example for all IDR mutants, see Fig. 6D). Time point 0 marks initial contact formation. **D** Quantification of the onset of cortical β-catenin enrichment after initial contact during de novo junction formation in β-catenin KO MDCK cells with mSc-β-catenin[Wt], mSc-β-catenin[IDRs*], mSc-β-catenin[nIDR*] or mSc-β-catenin[cIDR*] addback. Each point represents a measurement for a single de novo cell-cell contact ($n = 91$, 64, 55, and 73, respectively), pooled from three independent experiments (medians of each indicated with circles). **E** Two representative time-lapse images (Movies S17 and S18) exemplifying the unsuccessful formation of a linear cell-cell contact between two β-catenin KO MDCK cells with mSc-β-catenin[cIDR*] addback (Fire LUT), showing either the maintenance of dynamic protrusions rather than the linearization of the contact (top), or the failure to extend the contact and eventual junction breakage (bottom). Time point 0 marks initial contact formation. For all graphs P-values are indicated; normal (**B**) or nested (**D**) one-way ANOVA with Sidak's multiple comparison test. Scale bars represent 10 μm (overview images) or 1 μm (zoom-ins of cell-cell contacts). Source data are provided as a Source Data file.

model in which IDR-dependent clustering drives the assembly of dynamic E-cadherin/β-catenin clusters at the membrane. These clusters can mature into stable structures upon the formation of cell-cell contacts, potentially involving additional anchoring interactions that render the complex immobile.

### The formation of nascent cell-cell adhesions is promoted by IDR-dependent β-catenin clustering

Clustering of the cadherin complex increases the avidity of extracellular E-cadherin domains, which can enhance the probability of forming stable transmembrane interactions during the formation of de novo cell-cell adhesions[6–8]. Indeed, the formation of cadherin/catenin clusters has previously been observed during the formation of nascent cell-cell adhesions across different cell types, including epithelial Madin-Darby canine kidney (MDCK) cells[51–54]. To investigate the role of β-catenin IDR-dependent clustering during this process, we depleted endogenous β-catenin from MDCK cells and reintroduced either wildtype mSc-β-catenin or the different β-catenin-IDR mutants that are attenuated in their ability to form condensates (Figs. S6A, B and S6B). By following sparsely seeded single cells over time during the establishment of de novo cell-cell contacts, we observed the emergence of cortical β-catenin[Wt] in a clustered pattern within minutes after initial contact between protrusions of neighboring cells, similarly to what has been described previously for E-cadherin (Fig. 6A, B and Movies S11 and S15)[55]. In cells expressing either of the three β-catenin IDR mutants, the clustering of β-catenin in the initial phase of junction formation was significantly hampered. These mutant cells showed a strongly reduced presence or complete absence of cluster formation, with instead a more uniform appearance of β-catenin at the forming cell-cell contacts (Fig. 6A, B and Movies S12–14).

In wildtype cells, the emergence of β-catenin clusters is followed by lateral expansion of the new contact site, and the cell-cell contact reorganizes from dynamic protrusions into a linear junction in which β-catenin rearranges into a more uniform distribution (Figs. 6A–C and S6C and Movies S11 and S15). IDR-mutated cells were able to eventually establish cell-cell contacts after an extended period of time, indicated by the formation of a cohesive monolayer (Fig. S6C). However, mutant cells showed a significantly decreased efficiency of junction formation compared to wildtype cells. The rapid cortical accumulation of β-catenin to the novel contact site of two neighboring cells observed in wildtype cells was strongly delayed in all three mutant cell lines (Fig. 6C, D and Movies S12–14, S16). Furthermore, a significant fraction of cells expressing the β-catenin IDR mutants completely failed to form a linear cell-cell junction during the timespan of the experiment (3–4 h), with cells either maintaining dynamic protrusions without junction linearization or not extending the new contact and subsequently breaking their junction (Fig. 6B, E and Movies S17 and 18). Defects in junction formation in mutant cells

predominantly occurred between cell-cell contacts devoid of clustering (Fig. 6B), substantiating that IDR-dependent clustering of the cadherin complex increases the probability of successful cell-cell contact formation. Overall, these data demonstrate that β-catenin condensation facilitates clustering of the cadherin/catenin cell-cell adhesion complex, and this clustering contributes to the formation of novel sites of cell-cell adhesion.

## Discussion

In this study, combining in vitro characterizations and functional cellular analyses, we uncover a role for β-catenin condensation in the supramolecular organization of the cadherin complex (Fig. 7). We find that the disordered termini of β-catenin establish the formation of condensates that incorporate other components of the cadherin complex and nucleate the formation of submicron cadherin/catenin clusters. This β-catenin-dependent clustering is essential for the efficient formation of de novo cell-cell adhesions, demonstrating that the function of β-catenin in the cadherin complex extends beyond connecting cadherin to α-catenin and the actin cytoskeleton.

From this work and other recent studies, the formation of biomolecular condensates is emerging as a universal mechanism driving the assembly and clustering of cell adhesion complexes[19,20,28–31,56,57]. Across different adhesions, this involves the condensation of adaptor proteins that locally concentrate adhesion receptors and their interacting partners to accommodate their assembly into supramolecular structures. Our study demonstrates that β-catenin fulfils this role in the assembly of cadherin/catenin clusters (Fig. 7). As E-cadherin binds β-catenin after its synthesis and they traffic together to the plasma membrane[33,34], β-catenin and E-cadherin may already co-condense together at the membrane of secretory vesicles or shortly after arrival at the cell surface. This condensation may be aided by the dimensionality reduction of the membrane surface, which could increase the local concentration of β-catenin above the threshold required for condensate formation[58–61]. We hypothesize that these membrane-associated β-catenin biomolecular condensates mature from dynamic into solid structures capable of enduring forces. This is in line with β-catenin IDR-dependent clusters at cell-cell contacts of HCT116 cells, manifesting as immobile structures, whereas clusters at free membranes display more dynamic behavior. Condensate maturation is similarly proposed for other force-bearing biomolecular condensates, including FAs and centrosomes[62,63], resulting from stabilization of intrinsic interactions within the condensate and/or interactions with additional binding partners[64]. A potential candidate controlling the transition of E-cadherin-associated β-catenin condensates into immobile clusters is the actin cytoskeleton that rapidly connects to the cadherin/catenin complex after its arrival at the plasma membrane[33], akin to the actin-dependent stabilization of condensates of the TJ-component ZO[28]. The spatiotemporal kinetics of β-catenin

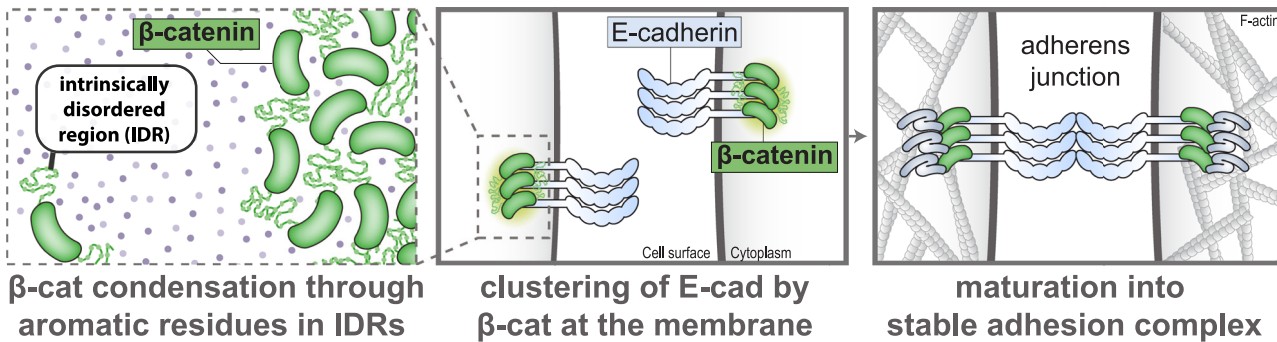

**Fig. 7 | Proposed model of β-catenin condensation-driven clustering of the cadherin/catenin complex.** β-Catenin forms biomolecular condensates through its N- and C-terminal intrinsically disordered regions (IDRs). β-Catenin retains this ability when associated with E-cadherin and thereby integrates E-cadherin into condensates. Consequently, β-catenin drives the formation of submicron clusters of E-cadherin/β-catenin at the cell cortex (or already at secretory vesicles). These β-catenin IDR-dependent clusters are essential for efficient formation of de novo cell- cell junctions and develop into sites of adhesion. The maturation of β-catenin IDR-dependent clusters into structures capable of enduring forces may involve linkage to the actin cytoskeleton or other interactions, which could potentially immobile the dynamic condensate. Clusters may eventually reorganize into a more uniform junction, as observed in MDCK cells following the completion of novel junction formation.

condensation in cells and the factors that regulate formation and maturation of condensates warrants further investigation.

We demonstrate that the condensate-forming properties of β-catenin are indispensable for clustering of the cadherin/catenin complex in both HCT116 cells and during nascent junction formation in MDCK cells. Clustering of the cadherin complex can enhance E-cadherin transdimerization by increasing the avidity of this interaction, which aids de novo junction formation as evidenced by theoretical modeling[6–8]. This role of cadherin complex clustering is supported by our experimental findings, showing defects in de novo cell-cell adhesions when IDR-dependent clustering is disrupted. β-Catenin-dependent condensation likely acts in concert with other clustering mechanisms of the cadherin complex, including *cis*-interactions between E-cadherin ectodomains, delimited diffusion by the actin cytoskeleton, and the involvement of other cadherin-associated proteins[9,10]. These alternative mechanisms may be preceded by cadherin/β-catenin condensation and immobilize and organize these clusters. In line with this, disruption of the actin cytoskeleton in HCT116 cells resulted in individual clusters coalescing into larger structures, suggesting that actin interactions may delimit cluster diffusion (Fig. S5C). Conversely, alternative clustering mechanisms may also promote condensate formation by locally concentrating cadherin/β-catenin complexes. The distinct clustering mechanisms may be partially redundant, and the contribution of individual mechanisms potentially diverges across cell types and species[9,10]. Future studies may resolve the interplay between the various clustering mechanisms of cadherin adhesions and how they coordinate the supramolecular organization of adherens junctions across different cell types.

The observation that β-catenin co-partitions E-cadherin into droplets in vitro is in line with structural analyses of the E-cadherin/β-catenin complex, which demonstrated that the high-affinity binding of β-catenin to E-cadherin is solely established by the Armadillo domain and does not involve the termini of β-catenin[40–42]. The IDRs are therefore expected to remain exposed in the E-cadherin/β-catenin heterodimer and able to form the interactions required to establish condensates. Even though a region of the N-terminal IDR of β-catenin contributes to α-catenin binding[42,65], we find α-catenin also to be incorporated into β-catenin droplets in vitro. Importantly, our functional analyses demonstrate that mutation of the C-terminal IDR of β-catenin is sufficient to attenuate E-cadherin/β-catenin clustering and junction formation, excluding any potential confounding effects on α-catenin association. Interestingly, we find that α-catenin also forms phase separated droplets by itself at similar concentrations as β-catenin, although it remains to be determined whether this contributes to

the organization of clusters of the cadherin complex (Fig. 1B). In contrast, our findings unambiguously demonstrate that β-catenin condensation is essential to drive the incorporation of all three core cadherin complex components together into droplets (Fig. 1) and submicron clusters at cell-cell contacts (Fig. 2).

The molecular composition of mature adherens junctions extends far beyond the core components described in this study[66], and these additional proteins may influence β-catenin condensates and vice versa. While disruption of β-catenin condensation is sufficient to attenuate clustering and junction formation, this does not exclude that other cadherin-associated proteins (in addition to α-catenin) may exhibit similar behavior and contribute to the condensation of the cadherin complex in cells. For instance, p120-catenin and Afadin, which are both implicated in clustering and adherens junction formation[67–71], both have predicted IDRs[72]. Conversely, in addition to cadherin clustering, β-catenin condensates may locally concentrate specific cadherin-complex interactors and thereby accelerate enzymatic reactions[10]. In this way, β-catenin condensation could impact signaling cascades or the activity of actin cytoskeletal regulators[20], analogous to increased actin nucleation at Nephrin adhesion receptors that is driven by Nck/N-WASP condensation[62]. How condensation of β-catenin acts together with other interaction partners of the cadherin/catenin complex to construct the supramolecular organization and function of adherens junctions warrants further investigation.

## Methods
### Antibodies
The following commercial antibodies were used at the indicated concentrations for Western blot (WB) and immunofluorescence (IF): rabbit anti-αE-catenin (Sigma-Aldrich; C2081; 1:500 IF); rabbit anti-β-catenin (Sigma; C2206; 1:2500 WB); mouse anti-β-catenin (BD Biosciences; 9018884; 1:1000 IF); mouse anti-α-tubulin (DM1A; Calbiochem; CP06; 1:5000 WB); rat anti-RFP (5F8; Chromotek; 1:2500 WB); and mouse anti-γ-catenin (Plakoglobin; Zymed; 13-8500 1:250 IF). For EM, biotin-anti-GFP (Rockland; 600-106-215; 1:300) and rabbit anti-biotin (Rockland; 100-4198; 1:10 000) antibodies were used.

### Plasmids
Plasmids for bacterial expression were generated by In-Fusion cloning of the indicated constructs into a pGEX-2T plasmid: mEGFP, mEGFP-β-catenin[Wt] (NM_001904.4;[36]), mEGFP-β-catenin[IDRs*36], mEGFP-β-catenin[nIDR*] and mEGFP-β-catenin[cIDR*], mTagBFP2, E-cadherin[cyto]-mTagBFP2 (E-cadherin NM_009864.2, amino acids 736–783; ref. 73) mCherry and mCherry-α-catenin (NM_009818.1; ref. 73). Lentiviral

plasmids were generated by In-Fusion cloning of the indicated constructs into a pLV-CMV plasmid: mScarlet-β-catenin[Wt] (NM_001904.4, ref. [36]), mScarlet-β-catenin[IDRs*36], mScarlet-β-catenin[nIDR*], mScarlet-β-catenin[cIDR*], E-cadherin[Δecto]-mScarlet (NM_001287125.2, ΔAA166-708), E-cadherin-mCherry (NM_001287125.2), mCherry-αE-catenin (NM_009818.1; ref. [74]), mCherry-(KRAS)CAAX (amino acids KMSKDGKKKKKKSKTKCVIM; ref. [75]) and mEGFP-α-catenin (NM_009818.1[76]). pcDNA3.1-Axin1b-mRFP1 (NM_181050.3) was a kind gift from the Maurice lab (UMC Utrecht, The Netherlands).

## Protein purifications

Recombinant proteins were expressed in the *E. coli* CK600K or BL21-DE3 Star strain and grown in Nutrient Broth n1 (Sigma-Aldrich). Following induction with 100 μM IPTG for 14–16 h at 25 °C, cells were harvested by centrifugation at 3470 × *g* for 30 min at 4 °C, resuspended in lysis buffer (50 mM Tris-HCl pH 7.5, 50 mM NaCl, 5 mM EDTA, 5 mM β-mercaptoethanol, and 5% glycerol), lysed by sonication followed centrifugation at 51428 × *g* for 30 min at 4 °C. Supernatants were loaded onto Glutathione Sepharose® 4B (Merck) beads in gravity columns (Bio-Rad). Columns were washed twice with 10 volumes of high salt washing buffer (50 mM Tris-HCl pH 7.5, 400 mM NaCl, 5 mM β-mercaptoethanol and 5% glycerol), once with 10 volumes of droplet formation buffer (50 mM Tris-HCl pH 7.5, 125 mM NaCl, 10% glycerol, 1 mM DTT) and once with TEV protease buffer (50 mM Tris−HCl pH 8.0, 0.5 mM EDTA and 1 mM DTT). TEV protease His-6 (Protean) cleavage was performed with 1.5 kU of the enzyme in 4 ml of TEV protease buffer overnight, and cleaved proteins were eluted with 2 volumes of TEV protease buffer. Proteins were further purified by gel filtration chromatography (Superdex 200 HiLoad 16/60 GE; Healthcare) in 20 mM Tris−HCl pH 8.0, 125 mM NaCl, and 1 mM DTT. For α-catenin, specifically, the monomeric pool was collected.

## Droplet assays

Recombinant proteins were diluted at indicated concentrations in droplet formation buffer (50 mM Tris-HCl pH 7.5, 125 mM NaCl, 10% glycerol, 1 mM DTT). After addition of 10% PEG-8000, the protein solution was immediately loaded onto microscope slides as a single drop. Slides were imaged with spinning disk confocal microscope (Nikon Ti2) with Apo TIRF 60× Oil DIC N2 lens or spinning disk CSU ×1 Nikon Ti with Plan Apo VC 60× N.A. 1.40 oil. Images were post-processed to correct for channel misalignment. Quantifications were performed according to a previously established pipeline[77], the code for this analysis is available at the following Github link: https://github.com/krishna-shrinivas/2020_Henninger_Oksuz_Shrinivas_RNA_feedback/tree/master/Droplet_analysis. Droplets were segmented by intensity, size and circularity thresholds and their intensity was calculated. The mean intensity of each droplet (C-in) and of the bulk (C-out) were calculated for each channel. The partition ratio was computed as (C-in)/(C-out), as a proxy for average per droplet concentration of fluorescent protein inside the condensates over the concentration of fluorescent protein outside the condensates. For each image, the total intensity of each droplet was calculated and then the sum was computed (TC-in) as well as the total bulk intensity of the whole field (TC-out). The condensed fraction was computed as (TC-in)/(TC-out), as a measure of the total amount of fluorescent protein inside the condensates over the total amount of fluorescent protein in a given image. Quantification data was filtered to exclude droplets with partition ratios lower than 1.1 for mCherry and mEGFP channels and lower than 1.35 for mTagBFP2 to correct for mEGFP bleedthrough. Two types of analysis were performed, one in which all the droplets for all the channels were considered for the condensed fraction (Figs. 1C and 3B) and one in which the droplet mask of β-catenin was used as a scaffold (β-catenin co-condensed fraction; Fig. 1G).

## Cell lines and culture

MDCK GII cells (gift from W.J. Nelson, Stanford University) were cultured at 37 °C and 5% CO$_2$ in low-glucose DMEM (Sigma; D5523-10L) supplemented with 10% FBS (Sigma; F7524-500ML), 1 × *g*/liter sodium bicarbonate, and Penicillin-Streptomycin (Lonza; LO DE17-602E). HCT116 (obtained from ATCC, CCL-247) and HEK293T (obtained from ATCC, CRL-3216) cells were cultured in high-glucose DMEM (Sigma; D6429-24X500ML) supplemented with 10% FBS and Penicillin-Streptomycin. V6.5 murine embryonic stem cells (gift from Rudolf Jaenisch, Whitehead Institute). were cultured at 37 °C and 5% CO$_2$ on 0.2% gelatinized tissue culture plates in 2i + LIF media, composed of DMEM/F12 (Lonza; BE04-687F/U1), N2 supplement (GIBCO; 17502048), B27 supplement (GIBCO; 17504001), 0.5 mM L-glutamine (Lonza; LO BE17-605E), 0.5× non-essential amino acids (Lonza; LO BE13-114E), Penicillin-Streptomycin, 0.1 mM β-mercaptoethanol (Sigma; M6250), 1 uM PD0325901 (Selleckchem; S1036), 3 uM CHIR99021 (Selleckchem; S2924), and 1000 U/mL recombinant LIF (Millipore; ESG1107). Live-cell imaging experiments were performed under the same culture conditions. All cell lines were regularly tested for the absence of mycoplasma.

Cell lines in which β-catenin is endogenously tagged with 3× mEGFP have been previously described[36]. Knock-out lines for β-catenin in HCT116 and MDCK cells were generated by CRISPR-Cas9, by transfection of pSpCas9(BB)−2A-GFP (Addgene #48138) with gRNA sequence (5′-GTTCCCACTCATACAGGACTTGG-3′) targeting human and canine β-catenin, using Lipofectamine LTX or Lipofectamine 3000, respectively. Two days following transfection, GFP-positive cells were sorted by FACS in 96-wells culture plates to obtain monoclonal lines. Clones were screened for successful knock-out of β-catenin by immunofluorescence, and validated by both western blotting and sequencing of the genomic locus.

Stable lines of HCT116 and MDCK cells expressing indicated constructs were generated using lentiviral transductions followed by selection with Puromycin (Bio Connect; SC-108071B) or Blasticidin (Bio Connect; ant-bl-1). In HCT116 and MDCK β-catenin knockout lines in which mSc-β-catenin[Wt] or mSc-β-catenin[IDRs*] were re-expressed, cells were sorted by FACS to obtain monoclonal lines. Plasmids E-cadherin[Δecto]-mScarlet and Axin1-mRFP1 were transiently transfected in HCT116 cells using Lipofectamine LTX and imaged two days later.

## Live-cell microscopy and analyses

For live-cell imaging, cells were seeded on glass-bottom dishes (Lab-Tek II), precoated with Rat Tail Collagen I (Corning). Cells were imaged on a Nikon Spinning Disc confocal microscope using a 60× objective (NA = 1.49). Imaging was performed at 37 °C and 5% CO$_2$ in temperature- and CO2-controlled incubators, using NIS-Elements software. To study the effect of actomyosin disruption, Cytochalasin D was added during image acquisition at a concentration of 2 μg/ml.

For the detection of β-catenin clusters at cell-cell contacts, lines (4 pixels wide) were manually drawn over cell-cell contacts with a clear linear morphology using ImageJ Fiji software[78]. Only cells with comparable levels of exogenous protein expression were included in the analyses. Line profiles were normalized to the 20th–40th lowest intensity values. Individual clusters were detected using BAR plugin Find Peaks[79], with a minimal peak amplitude of 50% (for endogenously tagged mEGFP-β-catenin) or 25% (for exogenously expressed mEGFP-β-catenin). For the quantification of number of peaks per μm, detected peaks were manually verified, blindly. The fraction of overlapping peaks was determined by counting the number of mEGFP peaks per cell-cell contact that had an mSc or mCherry peak detected within 3 pixels distance. The size of mEGFP-β-catenin clusters was determined by fitting a Gaussian function to line profiles of individual clusters and calculating the full-width at half-maximum ($FWHM = 2\sqrt{2\ln(2)}\,\sigma$) from the extrapolated standard deviation ($\sigma$).

For the observation of in vitro droplets fusion events, samples were prepared as explained in the droplet assays method and images were acquired every 3.16 s.

## Cell-cell junction formation

To analyze de novo junction formation, sub-confluent MDCK cells were trypsinized and sparsely seeded as single cells on collagen-coated glass-bottom dishes. After attachment and spreading of cells (~2 h), cells were imaged at a time interval of 2–3 min for 3–4 h, allowing single cells to migrate towards other cells and form new contacts. De novo contacts were manually scored for the duration from the first contact between cells until the start of cortical β-catenin signal appeared, for the presence of clusters during junction formation, and for the successful establishment of a linear junction (i.e., extension of the contact with no junction breakage or failure to reduce dynamic protrusions).

## Immunostainings

For immunostainings, confluent cell monolayers grown on collagen-coated glass-bottom dishes were fixed with 4% paraformaldehyde (PFA; Sigma-Aldrich); permeabilized with 0.2% Triton × -100 (Sigma-Aldrich); blocked in buffer containing 1% BSA (Sigma-Aldrich), 1% goat serum (Life Technologies), and 1% donkey serum (Jackson Immunoresearch); and incubated with the indicated primary and Alexa-conjugated secondary antibodies (Life Technologies), together with DAPI (Sigma-Aldrich) where indicated. Cells were analyzed on a Zeiss LSM880 scanning confocal microscope using a 63 × objective (NA = 1.15), using Zen image acquisition software.

## Fluorescent recovery after photobleaching (FRAP)

FRAP experiments were performed on a Zeiss LSM880 scanning confocal microscope using a 63 × objective (NA = 1.15), using Zen image acquisition software, or on a Nikon Spinning Disc confocal microscope with a FRAP module (OMS) using a 60 × objective (NA = 1.42), using NIS-Elements software. Imaging was performed at 37 °C and 5% $CO_2$ in temperature- and $CO_2$-controlled incubators. Individual clusters were bleached using 100% laser power and imaged over time (5–9.5 min) with a 10 s interval in a z-stack (5 × 0.5 μm interval). Fluorescent intensities were measured using ImageJ[78] by manually drawing ROIs over individual clusters and a concomitant distal region at the junction within maximum projections for each time frame. Background-subtracted florescent intensity values were normalized to the pre-bleaching and immediate post-bleach mean intensities. To calculate the mobile fraction ($F^{mob}$) and the half-life recovery time ($τ^{1/2}$), the average of 15 replicates was fitted to a single exponential curve in Prism 8 software (GraphPad):

$$Y(t) = Y(0) + \left(F^{mob} - Y(0)\right) * \left(1 - e^{-\frac{t\ln(2)}{τ^{1/2}}}\right)$$

## Correlative light electron microscopy (CLEM)

CLEM was performed as described in detail in refs. 80,81. HCT116 cells expressing endogenously tagged 3 × mEGFP-β-catenin were seeded on carbon- and collagen-coated coverslips and fixed in 2% PFA and 0.2% glutaraldehyde (GA). Cells were washed with PBS, incubated in PBS/0.15% glycine for 10 min, washed with PBS/0.1% BSA, and incubated with 12% gelatin (Merck; G1890) at 37 °C for 30 min. The 12% gelatin was then solidified at 4 °C for 30 min, and the coverslips were incubated with 2.3 M sucrose for 48 h at 4 °C. After incubation, the gelatin, including embedded cells, had spontaneously detached from the coverslip. The gelatin-embedded cell layer was then cut into blocks of ~1 mm³ and mounted on pins with cell bottoms facing up and snap-frozen and stored in liquid nitrogen.

Samples were cryosectioned into ribbons of 90–100 nm ultrathin sections and deposited on grids (Cell Microscopy Core, UMC Utrecht). The grids were incubated with PBS/0.15% glycine for 30 min at 37 °C, rinsed with PBS and incubated with 10 μg/mL DAPI in 0.1% BSA-c (Aurion; 900.099) and 0.5% fish skin gelatin (Sigma-Aldrich; G7765) in PBS (BSA-c/FSG) for 20 min. Grids were then washed 5× in PBS and 2× in 50% glycerol in 0.1 M PHEM (EMS; 11162) before being submerged in 50% glycerol in 0.1 M PHEM and sandwiched between a coverslip and glass slide. The sections were imaged for mEGFP signal and DAPI in this configuration on a Leica Thunder widefield microscope with 100× objective (NA = 1.47), Photometrics Prime 95B sCMOS camera and LAS X software. After image collection of the whole ribbon of sections, the grids were retrieved, washed 3× in PBS and incubated with BSA-c/FSG) for 10 min. Next, we labeled grids with biotin-anti-GFP antibody (Rockland; 600-106-215; 1:300) in BSA-c/FSG for 1 h. Grids were then washed 5× in PBS/0.1% BSA and incubated with bridging rabbit anti-biotin antibody (Rockland; 100-4198; 1:10000) in BSA-c/FSG for 20 min, followed by 5 more PBS/0.1% BSA washes and incubation with Protein-A conjugated to 10 nm gold particles (Cell Microscopy Core, UMC Utrecht; PAG10; 1:50) in BSA-c/FSG for 20 min. Grids were then washed 5× with PBS, incubated with 1% GA in 0.1 M phosphate buffer, 2× PBS, 8× milliQ, and contrasted using uranylacetate pH 7 (5 min) and uranylacetate:methylcellulose pH 4 (10 min, 4 °C). Excess uranyl was blotted away and grids were dried using the "loop-out" method[81]. The dried grids were imaged in a FEI Tecnai T12 transmission electron microscope (TEM) with Veleta VEL-FEI-TEC12-TEM camera and SerialEM software. Regions selected by the fluorescence images were correlated and imaged at 50,000× magnification in TEM. The obtained image tilesets were correlated based on DAPI signal and nuclear outlines using ImageJ Fiji software[78].

## Electron microscopy of resin-embedded cells

HCT116 3 × mEGFP-β-catenin cells (the same cell line as used in the CLEM experiments) were seeded in 3 cm dishes 24 h prior to fixation. Fixative consisted of 2.5% GA and 2% PFA in 0.1 M PHEM buffer and was added 1:1 to cell culture medium for 5 min at RT. Culture medium and fixative were then replaced with fixative alone for 2 h at RT. Samples were then stored in 0.5 PFA in 0.1 M PHEM at 4 °C. For contrasting, 1% OsO4 was used for 1 h. After a dehydration series in ethanol, samples were further processed in EPON in increasing concentrations and finally embedded in 100% EPON. EPON was polymerized for 5 d at 60 °C.

These resin-embedded samples were prepared for thin sectioning by removal from the culture dish, cutting out a rectangle of ~0.5 by 1 mm and mounting, bottom side up, on an EPON stub. We proceeded by cutting 60–70 nm sections on a Leica Ultracut S (Leica Microsystems) using a DiATOME Ultra Diamond Knife 45°. Sections were placed on Formvar- and carbon-coated copper grids and poststained in a Leica EM AC20 (Leica Microsystems) using uranyl and lead citrate. Samples were imaged on a Tecnai T20 (FEI Tecnai) TEM using Radius software.

## PONDR disorder score prediction

Disorder scores were calculated with the PONDR web tool: http://www.pondr.com.

## Statistics and reproducibility

All statistical analyses as indicated in the Figure legends, were performed using Prism 8 software (GraphPad) or R (version 4.5.1). Data were tested for normality, with two-sided tests employed unless otherwise specified, and corrections for multiple comparisons were made when indicated. No statistical method was used to predetermine sample size. Samples sizes are indicated in the legend of each experiment and were selected to provide adequate statistical power. For the selection of cell-cell contacts for analyses, only cells showing a proper

expression of the transfected constructs were included. For the comparison between wildtype and mutant proteins, we analyze cell-cell contacts showing comparable expression levels. Data were either analyzed using automated image analysis, or manually analyzed and under blinded conditions when possible.

## Reporting summary

Further information on research design is available in the Nature Portfolio Reporting Summary linked to this article.

## Data availability

Source data are provided with this paper. Original imaging data, plasmids and cell lines are available upon request. Source data are provided with this paper.

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

## Acknowledgements

We thank Ilya Grigoriev and Anna Akhmanova at Cell Biology, Utrecht University (Utrecht, The Netherlands) for the use of the spinning-disc confocal microscope for the droplet assay experiments, the Cell Microscopy Core UMC Utrecht for providing training, reagents, and equipment for the EM experiments, and members of our laboratories for helpful discussions. This work was supported by the Dutch Cancer Foundation (KWF-13589, J.S. and KWF-12345, M.G.) and the Netherlands Organization for Scientific Research (NWO; 016.Vidi.189.166, M.G., NWO gravitational program CancerGenomiCs.nl 024.001.028, M.G., and the Science-XL research program The Active Matter Physics of Collective Metastasis 2019.022, M.G.). The CLEM infrastructure used in this work is part of the Netherlands Electron Microscopy Infrastructure (NEMI), a National Roadmap program for Large-Scale Research Infrastructure, which is financed by the Dutch Research Council (project number 184.034.014, J.K.).

## Author contributions

J.L.M., J.S. and M.G. conceived the study. J.L.M., C.M., W.J.P. and J.A.H. performed experiments and analyzed data. J.A.B., C.H. and J.K. performed and supervised the EM experiments. M.H. assisted with the purification of recombinant proteins. M.G. supervised the study, and J.L.M. and M.G. wrote the manuscript with input from all authors.

## Competing interests

The authors declare no competing interests.
