## [Transparent Peer Review file · Nature Communications]

β -Catenin condensation facilitates clustering of the cadherin/catenin complex and formation of nascent cell-cell junctions

Corresponding Author: Dr Martijn Gloerich

Version 0:

Reviewer comments:

Reviewer #1

(Remarks to the Author)

In this manuscript, the authors summarized and discussed a novel clustering mechanism of the cadherin complex established by its core component β -catenin. On this basis, they further investigated the co-partition of β -catenin and other cadherin complex components, the IDR-dependent β -catenin clusters, and the importance of cadherin-complex clustering in establishing adhesion between two opposing cells. This work provides new insight and opinion into the function of β -catenin, which plays an essential role in the supramolecular organization of the cadherin complex. The manuscript is well-organized and clearly stated. The following issues need to be resolved before the article is published:

1. The authors should determine the material properties (viscosity) of the droplets in vitro and ideally compare them to nascent adhesions in cells, e.g. by FRAP or observing fusion events of droplets.
2. Page 7 Line 161. "similarly to β -catenin with both IDRs mutated (Figs. 3G and H)", there are no graph G and graph H in Figure 3. According to the content of the statement, it should be corrected as "similarly to β -catenin with both IDRs mutated (Figs. 2D and E)".
3. Page 7 Line 172. "fusion to a CAAX-motif...", there is a lack of explanation of what the CAAX-motif is.
4. In Fig. 4G, to avoid misunderstanding, it is suggested to replace "mEGFP- β -catenin + DNA" with "mEGFP- β -catenin + DAPI".
5. Page 9 Lines 204-206. The authors mention "We postulate that liquid-like assemblies of β -catenin guide initial cadherin complex clustering, and is followed by maturation of these clusters into a more stable state". The data are not enough to support this hypothesis, which should be demonstrated. Whether the dynamics of β -catenin differ with time or component composition, and incomplete fluorescence recovery after photobleaching may be attributed to plasma membrane regionalization rather than maturation of β -catenin clusters.

Reviewer #2

(Remarks to the Author)

The authors propose an interesting model in which beta catenin assembles into phase separated liquid droplets to promote the clustering of cadherin adhesion receptors. The authors describe solution studies with different proteins and with beta catenin mutants, and they then correlate the solution behavior with observed cadherin clustering at sites of adhesion between cells. The solution studies are noteworthy and there are some interesting findings.

The study makes some very strong claims about the role of beta catenin that are not convincingly supported by the data. The conclusion that liquid droplets drive cadherin clustering is also undermined somewhat by the FRAP data indicating that the puncta at cell adhesions are not in fact liquid like. The paper is also missing some key controls, and additional key studies are needed to rule out other proposed mechanisms. These points are described in more detail below.

If beta catenin undergoes liquid-liquid phase separation when associated with cadherin at the membrane, the process should be controlled entirely by the local protein concentration on the cytoplasmic face of the membrane. Phase separation is driven by concentrations, solubility, and the surface tension of the droplet. The physics should be independent of whether the cells are adhering or not. I did not see any evidence that these droplets form at free cell membranes. They seem to be

restricted to adhesions, suggesting that that other mechanisms are more important.

A related experiment would be to test whether proposed condensates form in cells, if the adhesion complexes are decoupled from the cytoskeleton. Does force or reduced cadherin mobility have any effect? If clustering is due to beta catenin phase separation driven by the local concentration, then force and mobility shouldn't have much effect. Along these lines, mutations that disrupt adhesion should not affect droplet formation.

Is the stoichiometry of components that form droplets in solution the same as at the membrane in the cell? Conversely, if the stoichiometric amounts of the components at the membrane are mixed in solution, do the proteins still form droplets in solution? This is needed to confirm that the same physics in solution is also operating in the cell.

Did they examine the effect of additives that disrupt phase segregated protein droplets, such as hexanediol?

What is the effect of extracellular domain cadherin mutants that disrupt lateral interactions? If their hypothesis that beta catenin plays a key role is correct, the intracellular phase segregation should be independent of weak interactions between the extracellular regions.

The proposed model is difficult to reconcile with the role of p120 catenin in clustering (Yap et al. 1998 J Cell Bio and Vu et al 2021 Curr Biol). If they eliminate p120 catenin, beta catenin should still bind, form droplets and cluster cadherins. This should be tested.

Reviewer #3

(Remarks to the Author)

Remarks to the Author:

This paper by Monster et al. addresses whether protein condensation of β -catenin is a prerequisite for clustering of proteins of the E-cadherin complex and subsequent formation of cell-cell junctions that mediate adhesion between cells in culture. Specifically, the authors use droplet formation assays and show that mutations in the intrinsically disordered N- and/or C-terminals of β -catenin inhibit the formation of phase separated droplets, that also can contain E-cadherin and α -catenin. By live-cell confocal microscopy they show that the same mutations that inhibit phase separation also inhibit the accumulation of cadherin/catenin complexes that form clusters along the cell-cell contact regions of the membrane in HCT116 cells. In addition, CLEM and FRAP analyses are used to determine the cluster's ultrastructure and stability, respectively. Lastly, live-cell confocal microscopy of sparsely seeded MDCK cells is used to visualize the dynamics of cluster organization during formation of cell-cell contact.

In general, I find this paper interesting, and most findings are both clear and well described. I have, however, suggestions and comments that might improve the manuscript.

Major points

1. Since β -catenin is part of adherens junctions, it is appropriate to analyze if there is an association between this type of cell-cell junction and individual mEGFP- β -catenin clusters. The authors address this by a CLEM analysis (Fig. 4G). Unfortunately, the fluorescence associated with membrane region shown is weak. It is similar to the weak intensity in regions between clusters (Fig. 4B, E) or alternatively, the weak fluorescence intensity before onset of β -catenin enrichment (Fig. 5C). Moreover, the EM resolution is low and the cell membranes with relative high electron density are separated with cell membranes that appear not to be intact. Together this makes it hard to agree with a conclusion that fluorescence intensity and electron density show an accumulation of β -catenin to adherens junctions. I would recommend that the authors show CLEM images of another region with intense mEGFP- β -catenin fluorescence. A suggestion is the intensely fluorescent cell-cell contact region located $\sim 10 \mu\text{m}$ to the left of the region boxed in yellow. The analyses should also include a statement about independent biological replicates and if possible, a reference to a paper showing electron micrographs of cell-cell junctions between HCT116 cells.

2. The authors combine CLEM with electron microscopy detection of mEGFP- β -catenin by protein-A-conjugated gold particles. The result of the gold particle detection does not show an association between β -catenin clusters or putative adherens junctions. This is due to the choice of analyzed region (commented above, pnt. 1) as well as an insufficient number of gold particles detected. As little as 5 cold particles are detected and only 3 of these appear to be located intracellularly and at proximity to the cell membrane. Accordingly, no quantification of immunogold staining is given that supports an increased accumulation β -catenin to the clusters or adherens junctions. This experiment should also be described in the result section and not only in M&M and figure legend.

Minor comments

3. I interpret the term "ectopic expression" to mean that a gene product is present at an abnormal location. Not that a gene is transcribed from a genetically modified and abnormally located genomic locus. However, I notice that the same terminology is used by Zamudio et al., who generated the endogenously tagged mEGFP- β -catenin (ref. 36). I therefore leave it to the editor to decide if terms "ectopic" and "ectopically", which are mentioned 22 times in the manuscript, are used correctly.

4. I don't understand why the authors make a distinction between stably integrated "ectopically" expressed transgenes and

the endogenously tagged mEGFP- β -catenin gene. The chimeric proteins generated by both types of transgenes are most likely not regulated or expressed in a manner that mimics their corresponding endogenous proteins during the dynamic events that establish cell-cell junctions. This issue is not addressed in the manuscript but is important for the discussion about which biomolecular interactions that are the driving force of cadherin cluster formation under physiological conditions (e.g. page 13, line 312)

5. The experiments shown in Fig. S4A lack a positive control, i.e. a figure showing that transiently expressed Axin1 is associated with clusters of β -catenin destruction complexes at the cell cortex. Perhaps the Axin1 expressed is sequestered into the cytosolic biomolecular condensates shown.

6. Page 7, line 161. β -catenin with both IDRs mutated (Figs. 3G and H). Should be (Figs. 2G and H).

7. Page 7, line 162-164. This sentence needs to be edited.

8. Fig. 2H, The unit on the Y-axis should be “.... per cell-cell contact” and not “.... per junction”.

Version 1:

Reviewer comments:

Reviewer #1

(Remarks to the Author)

All the comments and concerns have been cleared in the revised manuscript and can be accepted to be published.

Reviewer #2

(Remarks to the Author)

The authors have made several additions to the manuscript that strengthen their arguments. They have addressed many of the comments in the first review, and I commend them for carrying out several additional measurements that improve the data or address concerns. These additions improve the rigor of the study. There are still remaining concerns that are described below.

1. There is still one significant scientific weakness in this study. I have a scientific issue with the response to the role of p120ctn in clustering (Rev 2, comment 5). The authors state they are unaware of “experimental evidence supporting a defined role for p120-catenin (or other members of this protein family) in the formation of larger clusters of the cadherin-catenin complex”. In this context, they misinterpret Yap’s studies, which focused on the sequence N-terminal to the beta catenin binding region. Their comments suggest that p120 merely regulates cadherin at the membrane and they appear to be unaware that a dileucine mutation decouples internalization from p120ctn, protein surface expression and adhesion (Miyashita, Y., and Ozawa, M. (2007) Increased internalization of p120-uncoupled E-cadherin and a requirement for a dileucine motif in the cytoplasmic tail for endocytosis of the protein: <https://doi.org/10.1074/jbc.M608351200>). Also, see Miyashita, Y., and Ozawa, M. (2007) *J. Cell Sci.* 120, 4395–4406. They also appear to be unaware of the study by Vu et al *Curr Biol* (2022) that showed that p120ctn binding facilitates cadherin oligomerization, consistent with the Yap et al study. This omission is a significant weakness of the study. Although catenin may be involved in droplet formation (as their data suggest), the implication that the droplets direct larger cluster formation and the assembly of cadherin adhesions, independent of these other mechanisms, is not yet fully supported by their data.

2. In the rebuttal to Rev 2, the authors describe studies of droplets on free cell membranes. This is a positive addition to the paper and reveals differences between free and adhering complexes, presumably due to other factors contributing to adhesion assembly.

3. On pg 9, there is a suggestion that IDR clusters by themselves may enable the formation of stable structures at cell-cell contacts. However, once cells bind, cadherin self-organization and actin polymerization start kicking in and some of these processes can lead to stable oligomers in the absence of cytoplasmic domain interactions. A more physically plausible argument (based on these and prior data) is that the clusters form and then are further immobilized through actin anchorage and cis E-cadherin clustering at the contacts. In this context, the droplets aid the nucleation of cadherin clusters but are one of many participants in steps leading to “immobile” structures. Clarifying these points would strengthen the paper.

4. Also, “stable” could imply thermodynamic equilibrium, especially in this context of LLPS. The authors should discriminate between mechanical stability and thermodynamic stability.

5. The results showing the droplet coalescence after actin disruption is important and a nice addition.

6. In response to Rev 1 comments regarding comparisons of the physical properties (viscosity) of in vitro and in vitro (cells), they describe fusion studies. This does not actually address the reviewer. Fusion relates to surface tension, but viscosity reflects molecular organization in the droplets. The authors—to their credit—acknowledge that other proteins in cells may alter the behavior. However, this still begs the question regarding whether in vitro droplets are a model for what happens at the plasma membrane.

7. In response to Rev 1, comment 5, the authors altered their statement, but they still don’t meet the mark because they haven’t shown that the IDRs are driving the assembly of highly organized cadherin aggregates at cell adhesions. Here a broader, clearer model of what β -catenin is doing at different stages of adhesion assembly, in the context of other factors would be useful.

Reviewer #3

(Remarks to the Author)

My main critique was that it is difficult to agree that the CLEM analysis shows β -catenin fluorescence in adherens junctions between HCT116 cells. Since the β -catenin-associated plasma membrane is damaged, the β -catenin fluorescence could instead correspond to β -catenin fluorescence associated with plasma membrane fragments. The image from the same preparation (rebuttal Fig. 2) unfortunately shows that my concern was justified. To the right of inset 1 there is intense β -catenin staining in continuous plasma membrane regions that lack cell-cell contact. It is noteworthy that authors do not remove the initial image (Fig. 4G, now 5A) even though they agree that the plasma membrane is damaged (they also agree that the membranes are damaged in rebuttal Fig. 2). Instead, the fluorescence contrast in the original image now appears overly adjusted, which also is the case for an additional supplementary image with low EM resolution and damaged plasma membrane (Fig. S5A). The authors claim that the reason for the discontinuous plasma membrane is that a milder fixation is used in CLEM compared to the fixation used in traditional EM methods. While it is true that the antibody fluorescence is preserved by mild fixation, the fixation used (2% PFA, 0.2% glutaraldehyde) should not result in damage plasma membranes. Moreover, a putative explanation for poor membrane quality does not make the results more credible. The CLEM analysis was done with the aim to examine the ultrastructure of the β -catenin clusters. However, the results do not even reveal the ultrastructure of the electron-dense regions. The authors "mild fixation" explanation opens for a straightforward and simple experiment, which is to use a stronger fixation and perform a traditional EM. This should reveal the ultrastructure of electron-dense cell-cell contacts and if adherens junctions exist between HCT116 cells under the same culture conditions as for the CLEM analysis.

In accordance with my concern the conclusion about the immunogold staining is toned down in the revised version. The authors have also dealt with my minor comments appropriately except that they now get entangled in a reasoning about endogenous and exogenous gene expression that is confusing and as written, misses what is important. This is exemplified in the rebuttal letter where they write "endogenously tagged mEGFP- β -catenin knock-in utilizes the original promoter and regulatory elements, meaning its transcriptional regulation will mirror the native regulation of this protein". Of course, the transcriptional regulation will be mirrored, but not the regulation of the protein. It is not uncommon that a transcript and/or protein with a sequence corresponding to three consecutive copies of the green fluorescent proteins will interfere with translation and/or protein regulation (i.e. modifications, interactions, location and stability). Thus, it was the distinction between the endogenous (β -catenin) and exogenous (mEGFP- β -catenin) proteins I wanted the authors to pay attention to in their interpretation of the results, not the difference between exogenous gene transcription from knock-in loci versus stably transfected plasmids.

Reviewer #4

(Remarks to the Author)

Many key cell biological machines assemble from multivalent interactions, leading to formation of biomolecular condensates. Evidence now suggests that the cadherin-catenin complex at the adherens junction is one of them. One key component, beta-catenin, can phase separate in vitro, and evidence supports a role for this in its other cellular job, as part of the Wnt signaling pathway. Here the authors explore the role of beta-catenin phase separation in cell-cell adhesion. They find that the intrinsically disordered regions (IDRs) of beta-catenin play an important role in condensate formation in vitro and are important for efficient formation of junctional condensates with E-cadherin and alpha-catenin in cells, and they confirm a role for aromatic amino acids in this region. Condensates can form on non-contacting cell membranes, but trans-adhesion mediated by the E-cadherin extracellular domain increases their stability. Actin plays an interesting role, acting to prevent excessive condensate mergers. Finally, they provide evidence that the IDRs are important for timely and high fidelity initiation of cell-cell adhesion.

Since I was asked to come into this process after the original reviews and revision, I have confined my efforts to assessing whether the work is important, and the data support the conclusions, rather than suggesting new areas of investigation. I also have examined whether the authors have effectively addressed the concerns of the previous reviewers, particularly Reviewer 2 who was unavailable. I think they have effectively addressed all of the issues raised by Reviewer 2, some of which appear to have resulted from some confusion about points made in the original manuscript. They also appear to have addressed the issues raised by the other reviewers, though I leave it to them to comment on this. In summary, I think the authors work is exciting and well-supported and will be broadly read by cell, developmental and cancer biologists.

Minor issues that do not require re-review

1. I found the description of what is meant in Fig 1C by "normalized condensed fraction" incomplete. The data support the conclusion but a clearer explanation of what this metric precisely measures would be helpful. I also thought it worth mentioning that while the IDR** mutants reduced condensate formation, many of the condensates that did form in Fig 1F contained all three proteins.

2. The authors might note that cadherin puncta have been observed and molecules within them counted in vivo in *Drosophila* embryos. <https://pubmed.ncbi.nlm.nih.gov/19468069/>
<https://pubmed.ncbi.nlm.nih.gov/37163320/>

3. They might also note that in *Drosophila* the C-terminal IDR is important for Wnt signaling but is not essential for adhesion. Likewise, individually deleting the protein of the N-terminal IDR that is not involved in alpha-catenin binding also does not impair the role of beta-catenin in adhesion, suggesting potential redundancy in vivo.
<https://pubmed.ncbi.nlm.nih.gov/10471715/>

<https://pubmed.ncbi.nlm.nih.gov/9187151/>

Version 2:

Reviewer comments:

Reviewer #2

(Remarks to the Author)

Reviewer #3

(Remarks to the Author)

The revised manuscript now includes additional CLEM images and, as requested, high resolution EM images that complement the CLEM analysis (Fig. S5b). This together with additional citations and adequate description of the CLEM results have improved the conclusion made about the correlation between β -catenin clusters and the sites of intercellular adhesion. Thus, I consider that the revised version is suitable for publication in Nature communications.

We thank all Reviewers for their positive evaluation of our work and their constructive criticisms, which helped us to improve the manuscript. We have performed additional experiments and analyses, and revised the manuscript to address their criticisms, as detailed below in our point-to-point reply to their specific comments. We hope the Reviewers share our enthusiasm for our revised manuscript and find it suitable for publication in *Nature Communications*.

REVIEWER COMMENTS

Reviewer #1 (Remarks to the Author):

In this manuscript, the authors summarized and discussed a novel clustering mechanism of the cadherin complex established by its core component β -catenin. On this basis, they further investigated the co-partition of β -catenin and other cadherin complex components, the IDR-dependent β -catenin clusters, and the importance of cadherin-complex clustering in establishing adhesion between two opposing cells. This work provides new insight and opinion into the function of β -catenin, which plays an essential role in the supramolecular organization of the cadherin complex. The manuscript is well-organized and clearly stated.

The following issues need to be resolved before the article is published:

1. The authors should determine the material properties (viscosity) of the droplets *in vitro* and ideally compare them to nascent adhesions in cells, e.g. by FRAP or observing fusion events of droplets.

To address this point of the reviewer, we performed live-imaging of droplets containing β -catenin, E-cadherin, and α -catenin formed *in vitro*. This showed that these droplets underwent fusion events (Fig. 1E). Similarly, we found evidence in cells that IDR-dependent clusters of β -catenin at the cortex fused upon contact between clusters (Fig. S2C).

These new data have been included in the revised manuscript. Additionally, we acknowledge the limitations of comparing β -catenin-formed droplets *in vitro* with cellular clusters, as these conditions are not directly comparable (for example, cellular clusters likely contain additional components absent *in vitro*).

1E

S2C

coalescence of mEGFP- β -catenin clusters

2. Page 7 Line 161. “similarly to β -catenin with both IDRs mutated (Figs. 3G and H)”, there are no graph G and graph H in Figure 3. According to the content of the statement, it should be corrected as “similarly to β -catenin with both IDRs mutated (Figs. 2D and E)”.

We thank the Reviewer for pointing out this mistake and have now corrected it.

3. Page 7 Line 172. “fusion to a CAAX-motif...”, there is a lack of explanation of what the CAAX-motif is.

We acknowledge that the description of the CAAX motif required further clarification. A CAAX motif is a sequence consisting of a cysteine (C), two aliphatic amino acids (AA), and a terminal amino acid (X). It is found at the C-terminus of various proteins and facilitates their targeting to membranes. Specifically, we used the CAAX motif of K-Ras, which includes both this CAAX motif and an N-terminal polylysine sequence that directs K-Ras to the plasma membrane. The fusion of this motif to other proteins, such as the fluorescent protein mCherry in our case, is a commonly used strategy for plasma membrane targeting (e.g., <https://doi.org/10.1074/jbc.M007194200>).

To enhance clarity, we have revised the text and now introduce this as “*mCherry non-specifically targeted to the plasma membrane by fusion to a C-terminal plasma membrane localization signal (CAAX-motif of K-Ras, see Methods)*”. Additionally, in the Methods section, we provide further details and the exact sequence used.

4. In Fig. 4G, to avoid misunderstanding, it is suggested to replace “mEGFP- β -catenin + DNA” with “mEGFP- β -catenin + DAPI”.

We have adapted this in all Figures of the manuscript.

5. Page 9 Lines 204-206. The authors mention “We postulate that liquid-like assemblies of β -catenin guide initial cadherin complex clustering, and is followed by maturation of these clusters into a more stable state”. The data are not enough to support this hypothesis, which should be demonstrated. Whether the dynamics of β -catenin differ with time or component composition, and incomplete fluorescence recovery after photobleaching may be attributed to plasma membrane regionalization rather than maturation of β -catenin clusters.

We have extended our FRAP-based analysis of the IDR-dependent clusters at the cell cortex in HCT116 cells. In our original submission, we characterized the recovery kinetics after photobleaching specifically of clusters at cell-cell contacts. Based on Reviewer 2’s comments, we now demonstrate that IDR-dependent clusters also form at the free cell membrane (Figs. 2A and

S2E). This prompted us to re-examine our initial FRAP data and extend our analysis by comparing clusters at cell-cell contacts with those at the free membrane.

This analysis revealed that photobleached GFP- β -catenin clusters at cell-cell contacts did not show recovery above the fluorescence levels at the membrane outside of the clusters (Fig. 5D and S5B; and as was also seen in the Figure included in the original submission). In contrast, photobleached clusters at the free membrane show prominent fluorescent recovery within the cluster, with fast recovery kinetics ($t_{1/2} = 17s$) (Fig. 5E and F). These data imply that IDR-dependent clusters at the free membrane show more diffusivity of β -catenin, whereas this is reduced in clusters at cell-cell contacts.

These newly added FRAP data with comparison of the molecular dynamics of IDR-dependent clusters at the free membrane and cell-cell contacts does align with the model that initial IDR-dependent clustering drives the early assembly of E-cadherin/ β -catenin clusters at the membrane, which subsequently mature into a stable state after forming cell-cell contacts. Nonetheless, the reviewer's argument remains relevant, as other factors could influence differences in photorecovery kinetics. We therefore fully agree with this reviewer that we should be cautious in interpreting our findings. Therefore, we have toned down this statement and now describe that *“These findings indicate the existence of dynamic IDR-dependent E-cadherin/ β -catenin clusters at the cell cortex outside of cell-cell contacts. Altogether, our findings support a model in which IDR-dependent clustering drives the assembly of dynamic E-cadherin/ β -catenin clusters at the membrane, which can mature into a stable state upon forming cell-cell contacts.”*

Reviewer #2 (Remarks to the Author):

The authors propose an interesting model in which beta catenin assembles into phase separated liquid droplets to promote the clustering of cadherin adhesion receptors. The authors describe solution studies with different proteins and with beta catenin mutants, and they then correlate the solution behavior with observed cadherin clustering at sites of adhesion between cells. The solution studies are noteworthy and there are some interesting findings.

(*) The study makes some very strong claims about the role of beta catenin that are not convincingly supported by the data. The conclusion that liquid droplets drive cadherin clustering is also undermined somewhat by the FRAP data indicating that the puncta at cell adhesions are not in fact liquid like.

In our manuscript, we describe:

- i) that the IDRs of β -catenin drive the formation of phase-separated droplets in vitro and are essential for integrating E-cadherin and α -catenin into these droplets.
- ii) that the IDRs of β -catenin drive the formation of clusters at the cell cortex and are essential for integrating E-cadherin and α -catenin into these clusters.

The most parsimonious interpretation of these findings, in our view, is that β -catenin's ability to form IDR-dependent condensates contributes to the formation of β -catenin/E-cadherin clusters in cells. However, we acknowledge that we do not have direct evidence for the liquid-like behavior of cortical β -catenin clusters in cells. To prevent overinterpretation, we have revised the manuscript to further emphasize that our conclusions are based on a combination of in vitro and cellular experiments.

Additionally, in response to other comments from this Reviewer, we have expanded our FRAP-based analysis of IDR-dependent clusters at the cell cortex in HCT116 cells. In our original submission, we characterized the recovery kinetics after photobleaching specifically in clusters at cell-cell contacts. Following comment 1 of this Reviewer, we now demonstrate that IDR-dependent clusters also form at the free cell membrane (Figs. 2A and S2E). This prompted us to re-examine our initial FRAP data and extend our analysis by comparing clusters at cell-cell contacts with those at the free membrane. This analysis revealed that photobleached GFP- β -catenin clusters at cell-cell contacts did not recover beyond the fluorescence levels of the surrounding membrane outside of the clusters (Fig. 5D and S5B; and as was also seen in the Figure included in the original submission). In contrast, photobleached clusters at the free membrane exhibited prominent fluorescence recovery within the cluster, with fast recovery kinetics ($t_{1/2} = 17s$) (Fig. 5E and F). These results suggest that IDR-dependent clusters at the free membrane are more dynamic, while those at cell-cell contacts show reduced diffusivity. While we refrain from making claims about liquid-like behavior of these cellular clusters, our newly added FRAP data support the model in which IDR-dependent clustering drives the assembly of dynamic E-cadherin/ β -catenin clusters at the membrane, which can mature into a stable state upon forming cell-cell contacts.

(**) The paper is also missing some key controls, and additional key studies are needed to rule out other proposed mechanisms. These points are described in more detail below.

We address all specific comments related to this concern below. However, we would like to emphasize that *i*) our experiments in cells unambiguously demonstrate a role for the IDRs of β -catenin in organizing the β -catenin/E-cadherin complex into clusters and *ii*) this does not exclude the possibility that other mechanisms also contribute to the organization of this complex. In fact, we explicitly describe in our *Discussion* section (Page 13 line 15 – 27) that alternative mechanisms involved in clustering of the cadherin complex may be preceded by cadherin/ β -catenin condensation and further stabilize these clusters, or conversely, may promote condensate formation by locally concentrating cadherin/ β -catenin complexes. As discussed below, these additional mechanisms are not essential for the formation of IDR-dependent clusters. However, we believe it is important to consider that these mechanisms are interconnected and all are expected to influence cluster formation, as further outlined in our response to the specific comments.

1. If beta catenin undergoes liquid-liquid phase separation when associated with cadherin at the membrane, the process should be controlled entirely by the local protein concentration on the cytoplasmic face of the membrane. Phase separation is driven by concentrations, solubility, and the surface tension of the droplet. The physics should be independent of whether the cells are adhering or not. I did not see any evidence that these droplets form at free cell membranes. They seem to be restricted to adhesions, suggesting that that other mechanisms are more important.

We would like to note that trans-interactions of E-cadherin between cells will increase the retention of the E-cadherin/beta-catenin complex at the membrane (doi: 10.1083/jcb.142.4.1105) and thereby locally increase their concentration, which could thus promote condensation. Nonetheless, we agree that cell-cell adhesion should not be a strict requirement for this process. To support this, we have expanded our analysis to include β -catenin clusters at the free membrane, which were not shown in the Figures of our initial submission. Imaging of endogenously tagged β -catenin shows it forms clusters at the free membrane (Fig. 2A). Moreover, co-expression with mScarlet- β -catenin demonstrate that this clustering at the free cell membrane is dependent on the IDRs, similar to clusters at cell-cell contacts (Fig. S2E). We hope these additional data and discussion take away the Reviewer's concern.

2A**S2E**
2. A related experiment would be to test whether proposed condensates form in cells, if the adhesion complexes are decoupled from the cytoskeleton. Does force or reduced cadherin mobility have any effect? If clustering is due to beta catenin phase separation driven by the local concentration, then force and mobility shouldn't have much effect. Along these lines, mutations that disrupt adhesion should not affect droplet formation.

To the best of our knowledge, it is technically not feasible to genetically decouple the cadherin complex from the actin cytoskeleton while preserving the integrity of the cadherin-catenin complex. Therefore, the only viable approach to addressing this question is through chemical manipulation of the actin cytoskeleton. To this end, we treated HCT116 cells with Cytochalasin D, an inhibitor of actin polymerization. Under this condition, IDR-dependent clusters at the cell cortex remained present (**Fig. S5C**).

However, we observed that under following inhibition of actin polymerization, individual β -catenin clusters coalesced into larger structures. This suggests that the actin network, either through physical interactions or indirectly, is important for the dispersed distribution of β -catenin clusters.

S5C**disruption of the actin network**
Most importantly, these findings indicate that IDR-dependent clusters persist in the absence of actin. However, further understanding the functional interaction with the actin cytoskeleton provides additional layers and we respectfully argue that unraveling this interconnection is out of the scope of the current manuscript. Our primary conclusion remains that an IDR-dependent mechanism is essential for cluster formation.

We refer to our answer to comment #5 of this Reviewer for mutations that disrupt adhesion.

3. Is the stoichiometry of components that form droplets in solution the same as at the membrane in the cell? Conversely, if the stoichiometric amounts of the components at the membrane are mixed in solution, do the proteins still form droplets in solution? This is needed to confirm that the same physics in solution is also operating in the cell.

The stoichiometry of E-cadherin: β -catenin: α -catenin at the plasma membrane has been previously established as 1:1:1 (Ozawa and Kemler, 1992; Hinck et al., 1994). Based on this, we combined these proteins in our *in vitro* experiments at the same ratio. We now describe this in the manuscript.

p5, line 15-17: “However, when combined at equimolar levels, similar to their stoichiometry at the plasma membrane in cells^{33,39}, E-cadherin^{cyto}-mTagBFP2 incorporated into droplets of mEGFP- β -catenin (Figs. 1C and 1D).”

4. Did they examine the effect of additives that disrupt phase segregated protein droplets, such as hexanediol?

We analyzed the effect of hexanediol and found that it disrupts the majority of β -catenin clusters in HCT116 cells (Rebuttal Figure 1). However, we would prefer not to include these data in the manuscript. 1,6-Hexanediol has pleiotropic effects on condensates and more general on hydrophobic interactions throughout the cell, and is not very reliable to assess the presence of phase segregated proteins (PMID: 30682370). In our opinion, the IDR-mutated β -catenin proteins with reduced condensate-forming capacity represent a more precise experimental perturbation (Figure 2).

Rebuttal Figure 1

What is the effect of extracellular domain cadherin mutants that disrupt lateral interactions? If their hypothesis that beta catenin plays a key role is correct, the intracellular phase segregation should be independent of weak interactions between the extracellular regions.

To address this question, we introduced a truncated mutant of E-cadherin lacking almost the entirety of its extracellular domain (amino acids 166-708) into HCT116 cells. Despite this truncation disrupting lateral cis interactions between E-cadherin proteins, as well as homotypic trans interactions (thus disrupting the ability of E-cadherin to establish adhesion, see Comment #2), we found that this mutant still was able to cluster together with β -catenin at the free membrane. We included these data in the revised manuscript (Fig. S4A).

S4A

The proposed model is difficult to reconcile with the role of p120 catenin in clustering (Yap et al. 1998 J Cell Bio and Vu et al 2021 Curr Biol). If they eliminate p120 catenin, beta catenin should still bind, form droplets and cluster cadherins. This should be tested.

To the best of our knowledge, there is no experimental evidence supporting a defined role for p120-catenin (or other members of this protein family) in the formation of larger clusters of the cadherin-catenin complex. Instead, the role of p120-catenin in the organization of the cadherin complex is primarily linked to its regulation of cadherin stability at the plasma membrane by controlling cadherin endocytosis (doi: 10.1083/jcb.200307111); which explains the absence of cadherin clusters at the plasma membrane following the disruption of p120-catenin binding described in Yap et al., 1998). Thus, manipulating p120-catenin would primarily test how changes of E-cadherin/ β -catenin membrane levels, and thus local concentration, influence cluster formation. For this reason, we respectfully disagree on these experiments being essential to support our conclusion that β -catenin IDR's play a critical role in establishing E-cadherin/ β -catenin clusters.

Reviewer #3 (Remarks to the Author):

Remarks to the Author:

This paper by Monster et al. addresses whether protein condensation of β -catenin is a prerequisite for clustering of proteins of the E-cadherin complex and subsequent formation of cell-cell junctions that mediate adhesion between cells in culture. Specifically, the authors use droplet formation assays and show that mutations in the intrinsically disordered N- and/or C-terminals of β -catenin inhibit the formation of phase separated droplets, that also can contain E-cadherin and α -catenin. By live-cell confocal microscopy they show that the same mutations that inhibit phase separation also inhibit the accumulation of cadherin/catenin complexes that form clusters along the cell-cell contact regions of the membrane in HCT116 cells. In addition, CLEM and FRAP analyses are used to determine the cluster's ultrastructure and stability, respectively. Lastly, live-cell confocal microscopy of sparsely seeded MDCK cells is used to visualize the dynamics of cluster organization during formation of cell-cell contact.

In general, I find this paper interesting, and most findings are both clear and well described. I have, however, suggestions and comments that might improve the manuscript.

Major points

1. Since β -catenin is part of adherens junctions, it is appropriate to analyze if there is an association between this type of cell-cell junction and individual mEGFP- β -catenin clusters. The authors address this by a CLEM analysis (Fig. 4G). Unfortunately, the fluorescence associated with membrane region shown is weak. It is similar to the weak intensity in regions between clusters (Fig. 4B, E) or alternatively, the weak fluorescence intensity before onset of β -catenin enrichment (Fig. 5C). Moreover, the EM resolution is low and the cell membranes with relative high electron density are separated with cell membranes that appear not to be intact. Together this makes it hard to agree with a conclusion that fluorescence intensity and electron density show an accumulation of β -catenin to adherens junctions. I would recommend that the authors show CLEM images of another region with intense mEGFP- β -catenin fluorescence. A suggestion is the intensely fluorescent cell-cell contact region located $\sim 10 \mu\text{m}$ to the left of the region boxed in yellow. The analyses should also include a statement about independent biological replicates and if possible, a reference to a paper showing electron micrographs of cell-cell junctions between HCT116 cells.

We thank the reviewer for their suggestions to improve the manuscript and data presentation.

Regarding the fluorescence signal intensity in Fig. 4G compared to other images, it is important to stress that this panel shows fluorescence images of endogenously expressed mEGFP- β -catenin in 90-100 nm cryosections. The total amount of cell material is significantly less than that of whole cells imaged in other figures. This difference limits direct comparison, and we have now emphasized this in the *Results* section.

To preserve fluorescence, we also opted for mild fixation of the cells (0.2% glutaraldehyde + 2% formaldehyde) in contrast to traditional EM methods. This is why, as the reviewer noted, at some areas the plasma membrane and cell-cell junctions are not fully preserved. This was also the case for the specific region suggested by the reviewer, of which we included a CLEM image here for their interest (Rebuttal Figure 2). Importantly, we are convinced that this does not influence the interpretation of our data, as we still consistently observe sites of intimate cell-cell contact and electron-dense regions akin to the zonula adherens at regions marked by intense fluorescent spots of mEGFP- β -catenin. To emphasize this, we have included several additional examples from our CLEM analysis to further support our findings (Fig. S5A). Additionally, we now explicitly state that this analysis is based on 4 individual samples from 2 biological replicates.

Rebuttal figure 2

Additional CLEM figure of the area requested by the reviewer. Sample preparation as described in the main manuscript.

S5A

2. The authors combine CLEM with electron microscopy detection of mEGFP-β-catenin by protein-A-conjugated gold particles. The result of the gold particle detection does not show an association between β-catenin clusters or putative adherens junctions. This is due to the choice of analyzed region (commented above, pnt. 1) as well as an insufficient number of gold particles detected. As little as 5 gold particles are detected and only 3 of these appear to be located intracellularly and at proximity to the cell membrane. Accordingly, no quantification of immunogold staining is given that supports an increased accumulation β-catenin to the clusters or adherens junctions. This experiment should also be described in the result section and not only in M&M and figure legend.

We acknowledge that the inclusion of protein-A-conjugated gold particles in our images required further clarification.

Indeed, only few protein-A-conjugated gold particles are present in the example image. The low amount of mEGFP- β -catenin that can be detected (as outlined in our answer to the previous comment), combined with the relative inefficiency of gold labeling compared to fluorescence, explains why not all fluorescent spots correspond to an abundance of gold particles in the EM images. This indicates that gold labeling is not optimal for visualizing mEGFP- β -catenin clusters, reinforcing our rationale for using CLEM to localize mEGFP- β -catenin in a more sensitive manner. While gold particles remain present in our CLEM analysis, they are thus not essential for detecting β -catenin clusters but merely provide localization at higher-resolution, albeit for only a subset of the available mEGFP- β -catenin molecules.

We now describe the rationale of initially including gold particles and our interpretation more clearly in the *Results* section when introducing the EM analysis.

Minor comments

3. I interpret the term “ectopic expression” to mean that a gene product is present at an abnormal location. Not that a gene is transcribed from a genetically modified and abnormally located genomic locus. However, I notice that the same terminology is used by Zamudio et al., who generated the endogenously tagged mEGFP- β -catenin (ref. 36). I therefore leave it to the editor to decide if terms “ectopic” and “ectopically”, which are mentioned 22 times in the manuscript, are used correctly.

We have now adjusted this wording and instead use the term ‘exogenous’ to indicate that these proteins were not endogenously expressed but exogenously introduced into the cells.

4. I don’t understand why the authors make a distinction between stably integrated “ectopically” expressed transgenes and the endogenously tagged mEGFP- β -catenin gene. The chimeric proteins generated by both types of transgenes are most likely not regulated or expressed in a manner that mimics their corresponding endogenous proteins during the dynamic events that establish cell-cell junctions. This issue is not addressed in the manuscript but is important for the discussion about which biomolecular interactions that are the driving force of cadherin cluster formation under physiological conditions (e.g. page 13, line 312)

We make this distinction solely to accurately describe how the expression was technically achieved in each experiment.

We would like to highlight that the endogenously tagged mEGFP- β -catenin knock-in utilizes the original promoter and regulatory elements, meaning its transcriptional regulation will mirror the native regulation of this protein. In contrast, exogenously expressed proteins are randomly integrated and driven by a synthetic promoter. However, we do not make any further distinction or claim differences in the behavior or regulation of endogenously versus exogenously expressed β -catenin.

5. The experiments shown in Fig. S4A lack a positive control, i.e. a figure showing that transiently expressed Axin1 is associated with clusters of β -catenin destruction complexes at the cell cortex. Perhaps the Axin1 expressed is sequestered into the cytosolic biomolecular condensates shown.

We apologize for the interpretation of this experiment not being fully clear. The purpose of this experiment was to rule out the possibility that β -catenin clusters at the cell cortex represent the destruction complex, and thus represent Axin1-containing clusters. If this were the case, we would expect exogenously expressed Axin1 to be present in these clusters as well. We do not

observe Axin1 in cortical clusters; despite the clear presence of β -catenin clusters at the membrane, this does not overlap with the distribution of Axin1. Thus, this indicates that β -catenin clusters do not represent clusters containing components of the destruction complex. There is thus also no positive control for their combined presence at the membrane, as our results show they do not reside together in membrane-associated clusters. We have rephrased the text referring to these data for clarity.

6. Page 7, line 161. β -catenin with both IDRs mutated (Figs. 3G and H). Should be (Figs. 2G and H).

We thank the Reviewer for pointing out this mistake and have corrected this.

7. Page 7, line 162-164. This sentence needs to be edited.

We have rephrased this sentence for clarity.

8. Fig. 2H, The unit on the Y-axis should be “.... per cell-cell contact” and not “.... per junction”.

We have adjusted this throughout the Figures of the manuscript.

We thank all reviewers for their continued investment in our work and constructive feedback. We have made textual adaptations in the manuscript and performed an additional EM experiment to address their remaining concerns, as detailed below in our point-to-point reply to their specific comments. We hope the Reviewers share our enthusiasm for our revised manuscript and find it suitable for publication in Nature Communications.

Reviewer #1 (Remarks to the Author):

All the comments and concerns have been cleared in the revised manuscript and can be accepted to be published.

Response:

We thank the Reviewer for supporting the publication of our manuscript.

Reviewer #2 (Remarks to the Author):

The authors have made several additions to the manuscript that strengthen their arguments. They have addressed many of the comments in the first review, and I commend them for carrying out several additional measurements that improve the data or address concerns. These additions improve the rigor of the study. There are still remaining concerns that are described below.

1. There is still one significant scientific weakness in this study. I have a scientific issue with the response to the role of p120ctn in clustering (Rev 2, comment 5). The authors state they are unaware of “experimental evidence supporting a defined role for p120-catenin (or other members of this protein family) in the formation of larger clusters of the cadherin-catenin complex”. In this context, they misinterpret Yap’s studies, which focused on the sequence N-terminal to the beta catenin binding region. Their comments suggest that p120 merely regulates cadherin at the membrane and they appear to be unaware that a dileucine mutation decouples internalization from p120ctn, protein surface expression and adhesion (Miyashita, Y., and Ozawa, M. (2007) Increased internalization of p120-uncoupled E-cadherin and a requirement for a dileucine motif in the cytoplasmic tail for endocytosis of the protein: <https://doi.org/10.1074/jbc.M608351200>). Also, see Miyashita, Y., and Ozawa, M. (2007) J. Cell Sci. 120, 4395–4406. They also appear to be unaware of the study by Vu et al Curr Biol (2022) that showed that p120ctn binding facilitates cadherin oligomerization, consistent with the Yap et al study. This omission is a significant weakness of the study. Although b-catenin may be involved in droplet formation (as their data suggest), the implication that the droplets direct larger cluster formation and the assembly of cadherin adhesions, independent of these other mechanisms, is not yet fully supported by their data.

Response:

We thank the reviewer for highlighting these additional studies. While, to our understanding, Vu et al. focus on dimerization rather than the formation of larger clusters *per se*, we agree that these findings related to p120-catenin are relevant and important to incorporate in our manuscript. Accordingly, we have made the following revisions:

1. In the final paragraph of the Discussion, where we had already noted a role for p120-catenin in clustering, we have now included all references suggested by the reviewer:

Lines 351-353: *“For instance, p120-catenin and Afadin, which are both implicated in clustering and adherens junction formation⁶⁷⁻⁷¹, both have predicted IDRs⁷².”*

2. In the 3rd paragraph of the *Discussion*, we discussed how β -catenin condensation likely acts in concert with other clustering mechanisms of the cadherin complex. We now highlight that besides *cis*-interactions between E-cadherin ectodomains and delimited diffusion by the actin cytoskeleton, also the involvement of other cadherin-associated proteins represents another mechanism of clustering:

Lines 317-320: “ *β -Catenin-dependent condensation likely acts in concert with other clustering mechanisms of the cadherin complex, including cis-interactions between E-cadherin ectodomains, and delimited diffusion by the actin cytoskeleton, and the involvement of other cadherin-associated proteins.*”

Finally, we would like to reiterate that we do not conclude from our data that β -catenin condensation alone drives the formation of larger clusters or the assembly of cadherin adhesions. We elaborately discuss this in paragraph 3 and 5 of the *Discussion*, and further refer to our answer to Comment #3 of this Reviewer for this.

2. In the rebuttal to Rev 2, the authors describe studies of droplets on free cell membranes. This is a positive addition to the paper and reveals differences between free and adhering complexes, presumably due to other factors contributing to adhesion assembly.

Response:

We thank the Reviewer for positively evaluating these newly added data.

3. On pg 9, there is a suggestion that IDR clusters by themselves may enable the formation of stable structures at cell-cell contacts. However, once cells bind, cadherin self-organization and actin polymerization start kicking in and some of these processes can lead to stable oligomers in the absence of cytoplasmic domain interactions. A more physically plausible argument (based on these and prior data) is that the clusters form and then are further immobilized through actin anchorage and *cis* E-cadherin clustering at the contacts. In this context, the droplets aid the nucleation of cadherin clusters but are one of many participants in steps leading to “immobile” structures. Clarifying these points would strengthen the paper.

Response:

We fully agree with the reviewer's interpretation of our data. In line with this, in the 3rd paragraph of the *Discussion*, we discuss that β -catenin condensation may nucleate clusters, but that other mechanisms of clustering will likely contribute to the immobilization and further organization of these clusters. However, we acknowledge that this model may not have been described well enough throughout the manuscript. To ensure this point is unambiguous, we have also clarified it in the legend of our model figure (Figure 7), and now explicitly address it at the first point of data interpretation in the Results section:

Lines 233–237: “*Altogether, our findings support a model in which IDR-dependent clustering drives the assembly of dynamic E-cadherin/ β -catenin clusters at the membrane. These clusters can mature into stable structures upon the formation of cell-cell contacts, potentially involving additional anchoring interactions that render the complex immobile.*”

Finally, in line with the reviewer's suggestion, we now use the term *nucleation* to describe how β -catenin condensation contributes to catenin/cadherin cluster formation.

4. Also, “stable” could imply thermodynamic equilibrium, especially in this context of LLPS. The authors should discriminate between mechanical stability and thermodynamic stability.

Response:

When first introducing the term ‘stable clusters’, we now clarify that we mean *immobile* clusters. Where appropriate, we have also replaced the term *stable* with *immobile* to avoid confusion that we aim to discriminate between mechanical and thermodynamic stability.

Lines 226–229: “This showed that mEGFP- β -catenin clusters at cell-cell contacts did not recover beyond the fluorescence levels of the surrounding membrane outside of the clusters, indicating that the observed IDR-dependent clusters at cell-cell contacts are immobile, stable structures (Fig. 5D, S5C and Movie S9).”

5. The results showing the droplet coalescence after actin disruption is important and a nice addition.

Response:

We thank the Reviewer for positively evaluating these newly added data

6. In response to Rev 1 comments regarding comparisons of the physical properties (viscosity) of *in vitro* and *in vivo* (cells), they describe fusion studies. This does not actually address the reviewer. Fusion relates to surface tension, but viscosity reflects molecular organization in the droplets. The authors—to their credit—acknowledge that other proteins in cells may alter the behavior. However, this still begs the question regarding whether *in vitro* droplets are a model for what happens at the plasma membrane.

Response:

Reviewer 1 asked us to test material properties, specifically requesting an analysis of droplet fusion events. We conducted these experiments accordingly to the satisfaction of this reviewer. However, we remain careful in our interpretation of these data in the manuscript and do not make any claims regarding material properties or viscosity.

Regarding the validity of the *in vitro* model, as stated in our original response to this comment, we fully agree that direct comparisons between the properties of *in vitro* condensates and those formed *in vivo* are not possible. While *in vitro* droplets cannot fully recapitulate cluster formation in cells, we maintain that this method represents an effective and appropriate approach for studying protein condensation properties, precisely because it offers a minimalistic and controlled system.

7. In response to Rev 1, comment 5, the authors altered their statement, but they still don't meet the mark because they haven't shown that the IDRs are driving the assembly of highly organized cadherin aggregates at cell adhesions. Here a broader, clearer model of what β -catenin is doing at different stages of adhesion assembly, in the context of other factors would be useful.

Response:

Please see our answer to comment #3 of this Reviewer

Reviewer #3 (Remarks to the Author):

My main critique was that it is difficult to agree that the CLEM analysis shows β -catenin fluorescence in adherens junctions between HCT116 cells. Since the β -catenin-associated plasma membrane is damaged, the β -catenin fluorescence could instead correspond to β -catenin fluorescence associated with plasma membrane fragments. The image from the same preparation (rebuttal Fig. 2) unfortunately shows that my concern was justified. To the right of inset 1 there is intense β -catenin staining in continuous plasma membrane regions that lack cell-cell contact. It is noteworthy that authors do not remove the initial image (Fig. 4G, now 5A) even though they agree that the plasma membrane is damaged (they also agree that the membranes are damaged in rebuttal Fig. 2). Instead, the fluorescence contrast in the original image now appears overly adjusted, which also is the case for an additional supplementary image with low EM resolution and damaged plasma membrane (Fig. S5A). The authors claim that the reason for the discontinuous plasma membrane is that a milder fixation is used in CLEM compared to the fixation used in traditional EM methods. While it is true that the antibody fluorescence is preserved by mild fixation, the fixation used (2% PFA, 0.2% glutaraldehyde) should not result in damage plasma membranes. Moreover, a putative explanation for poor membrane quality does not make the results more credible. The CLEM analysis was done with the aim to examine the ultrastructure of the β -catenin clusters. However, the results

do not even reveal the ultrastructure of the electron-dense regions. The authors “mild fixation” explanation opens for a straightforward and simple experiment, which is to use a stronger fixation and perform a traditional EM. This should reveal the ultrastructure of electron-dense cell-cell contacts and if adherens junctions exist between HCT116 cells under the same culture conditions as for the CLEM analysis.

Response:

We acknowledge that the plasma membrane quality of our previously included Supplementary Figure was suboptimal, we therefore have taken additional images from our CLEM samples and included novel examples (Fig. S5A). Furthermore, as requested, we have performed traditional Resin-embedded EM to benchmark the structure of cell-cell junctions in HCT116 cells and included these data in the manuscript (Fig. S5B).

Importantly, the overall morphology of cell-cell contacts between these cells appears identical in the Resin-embedded sections and similar to other studies (PMIDs: 30863155, 26316041, 31215349). The cells are spaced relatively far apart and form only sparse cell-cell adhesions of a focal nature. Both datasets contain two morphologically distinct adhesions in HCT116 cells (Fig. S5A,B). One is characterized by prominent, electron-dense bands below the plasma membrane, typical of desmosomes, and where no fluorescent β -catenin enrichment is detected in the CLEM experiments. The other represents clear intercellular attachments with less pronounced intracellular protein density, resembling adherens junctions (albeit predominantly focal instead of linear) and representing the β -catenin-positive sites observed in CLEM.

In addition, in both experiments we find multiple examples of ruptured junctions, indicating that this is a biological phenomenon rather than a technical artifact. In line with this, we also observed breakage of cell-cell junctions between HCT116 cells with live-cell imaging. We occasionally also observed somewhat discontinuous plasma membranes, which likely originate from non-orthogonal sectioning as is more commonly observed with EM preparations, however these membranes were not associated with mEGFP enrichment. While we find more technical artifacts in CLEM (e.g tissue folds as in the original Rebuttal Fig. 2), these regions have been excluded from further analysis.

Importantly, in well-preserved regions, we consistently find that fluorescent β -catenin clusters overlap with electron-dense regions with adjacent membranes in close proximity. We therefore remain confident that our conclusion that fluorescent β -catenin clusters correspond to regions of cell-cell adhesion is well supported, and not confounded by technical limitations of CLEM. Furthermore, this interpretation that β -catenin puncta correspond to adhesion sites does not rely solely on the CLEM data (Fig. 5A), as this is further supported by the subsequent mosaic culture experiments (Fig. 5B–D).

Nonetheless, we acknowledge that it would be good to moderate the description of the CLEM findings, i.e., avoiding language that implies we are characterizing the ultrastructure of all beta-catenin puncta, removing the direct comparison to the zonula adherens, and describing the focal nature of the observed adhesions. Accordingly, we have made the following revisions:

Lines 201, 209-214: “*To further examine IDR-dependent β -catenin clusters, we applied electron microscopy (EM). ... By overlaying the two images we discerned that clusters of mEGFP- β -catenin identified by light microscopy represent cell-cell junctions, typically of a focal nature, with high protein densities at the adjacent cytosolic sites (Figs. 5A and S5A). The overall cell morphology and of the identified cell-cell adhesions was similar in classical resin-embedded EM, validating our CLEM findings (Fig. S5B)⁴⁸⁻⁵⁰. These results indicate that the IDR-dependent β -catenin clusters at cell-cell contacts represent sites of intercellular adhesion.*”

Supplementary Figure 5

In accordance with my concern the conclusion about the immunogold staining is toned down in the revised version. The authors have also dealt with my minor comments appropriately except that they now get entangled in a reasoning about endogenous and exogenous gene expression that is confusing and as written, misses what is important. This is exemplified in the rebuttal letter where they write “endogenously tagged mEGFP- β -catenin knock-in utilizes the original promoter and regulatory elements, meaning its transcriptional regulation will mirror the native regulation of this protein”. Of course, the transcriptional regulation will be mirrored, but not the regulation of the protein. It is not uncommon that a transcript and/or protein with a sequence corresponding to three consecutive copies of the green fluorescent proteins will interfere with translation and/or protein regulation (i.e. modifications, interactions, location and stability). Thus, it was the distinction between the endogenous (β -catenin) and exogenous (mEGFP- β -catenin) proteins I wanted the authors to pay attention to in their interpretation of the results, not the difference between exogenous gene transcription from knock-in loci versus stably transfected plasmids.

Response:

We apologize for misinterpreting the initial Reviewer’s comment about the difference between endogenous and exogenous gene tagging. We agree that we cannot exclude that adding a tag to a protein may impact its properties, which is an unavoidable limitation of all techniques involving live-imaging of proteins. We want to stress that we attempted to minimize this risk by using long linkers between β -catenin and the fluorescent tags, and that in all our experimental conditions we compare conditions where in both situations the same tag is present (e.g., cluster formation is only observed in wildtype beta-catenin and not IDR mutants, despite having the same tag). Altogether we are confident that the presence of fluorescent tags did not affect our overall conclusions.

Reviewer #4 (Remarks to the Author):

Many key cell biological machines assemble from multivalent interactions, leading to formation of biomolecular condensates. Evidence now suggests that the cadherin-catenin complex at the adherens junction is one of them. One key component, beta-catenin, can phase separate in vitro, and evidence supports a role for this in its other cellular job, as part of the Wnt signaling pathway. Here the authors explore the role of beta-catenin phase separation in cell-cell adhesion. They find that the intrinsically disordered regions (IDRs) of beta-catenin play an important role in condensate formation in vitro and are important for efficient formation of junctional condensates with E-cadherin and alpha-catenin in cells, and they confirm a role for aromatic amino acids in this region. Condensates can form on non-contacting cell membranes, but trans-adhesion mediated by the E-cadherin extracellular domain increases their stability. Actin plays an interesting role, acting to prevent excessive condensate mergers. Finally, they provide evidence that the IDRs are important for timely and high-fidelity initiation of cell-cell adhesion.

Since I was asked to come into this process after the original reviews and revision, I have confined my efforts to assessing whether the work is important, and the data support the conclusions, rather than suggesting new areas of investigation. I also have examined whether the authors have effectively addressed the concerns of the previous reviewers, particularly Reviewer 2 who was unavailable. I think they have effectively addressed all of the issues raised by Reviewer 2, some of which appear to have resulted from some confusion about points made in the original manuscript. They also appear to have addressed the issues raised by the other reviewers, though I leave it to them to comment on this. In summary, I think the authors work is exciting and well-supported and will be broadly read by cell, developmental and cancer biologists.

Minor issues that do not require re-review

1. I found the description of what is meant in Fig 1C by “normalized condensed fraction” incomplete. The data support the conclusion but a clearer explanation of what this metric precisely measures would be helpful.

Response:

We apologize for the lack of clarity regarding the normalized condensed fraction measurement. Although we had briefly explained this measure in the Figure legends and described in the *Methods* section how we calculated and normalized this condensed fraction, we have now expanded the explanation in the Methods to provide greater clarity.

I also thought it worth mentioning that while the IDR** mutants reduced condensate formation, many of the condensates that did form in Fig 1F contained all three proteins.

Response:

We thank the Reviewer for pointing this out, and we now included a statement about this when describing these results:

Lines 126-128: *“The small fraction of β -catenin^{IDRs*} droplets that did form contained all three proteins, indicating that the mutations impair the ability of β -catenin to form droplets but not its interactions with E-cadherin and α -catenin (Fig. 1G).”*

2. The authors might note that cadherin puncta have been observed and molecules within them counted in vivo in *Drosophila* embryos. <https://pubmed.ncbi.nlm.nih.gov/19468069/>
<https://pubmed.ncbi.nlm.nih.gov/37163320/>

Response:

We thank the reviewer for referring us to these articles. However, as we cite general Reviews when discussing the various types of cadherin puncta observed across model systems, we have not incorporated these specific references into the manuscript.

3. They might also note that in *Drosophila* the C-terminal IDR is important for Wnt signaling but is not essential for adhesion. Likewise, individually deleting the protein of the N-terminal IDR that is not involved in alpha-catenin binding also does not impair the role of beta-catenin in adhesion, suggesting potential redundancy in vivo. <https://pubmed.ncbi.nlm.nih.gov/10471715/>
<https://pubmed.ncbi.nlm.nih.gov/9187151/>

Response

We thank the Reviewer for referring us to these articles, we now cite this work when discussing the mutations introduced in the IDRs:

Lines 121-124: *“We previously identified that condensate formation of β -catenin relies on weak intermolecular interactions through aromatic amino acids in its N- and C-terminal IDRs³⁶⁻³⁸, which do not overlap with the established high-affinity binding sites in β -catenin for E-cadherin (Fig. S1D)⁴⁰⁻⁴⁴.”*